# Climate change alters low flows in Europe under a global warming of 1.5, 2, and 3 degrees

Marx Andreas[1], Kumar Rohini[1], Thober Stephan[1], Rakovec Oldrich[1,5], Wanders Niko[2,3], Zink Matthias[1], Wood Eric F.[3], Pan Ming[3], Sheffield Justin[4], and Samaniego Luis[1]

[1]Department Computational Hydrosystems, Helmholtz Centre for Environmental Research (UFZ), Leipzig, Germany
[2]Department of Physical Geography, Faculty of Geosciences, University Utrecht, The Netherlands
[3]Department of Civil and Environmental Engineering, Princeton University, Princeton, NJ, United States
[4]Geography and Environment, University of Southampton, Southampton, United Kingdom
[5]Faculty of Environmental Sciences, Czech University of Life Sciences, Czech Republic

*Correspondence to:* Andreas Marx (klima@ufz.de)

**Abstract.** There is growing evidence that climate change will alter water availability in Europe. Here, we investigate how hydrological low flows are affected under different levels of future global warming (i.e., 1.5, 2 and 3 K with respect to the pre-industrial period) in rivers with a contributing area of more than 1000 km$^2$. The analysis is based on a multi-model ensemble of 45 hydrological simulations based on three representative concentration pathways (RCP2.6, RCP6.0, RCP8.5), five Coupled Model Intercomparison Project Phase 5 (CMIP5) general circulation models (GCMs: GFDL-ESM2M, HadGEM2-ES, IPSL-CM5A-LR, MIROC-ESM-CHEM, NorESM1-M) and three state-of-the-art hydrological models (HMs: mHM, Noah-MP, and PCR-GLOBWB). High resolution model results are available at a spatial resolution of 5 km across the Pan-European domain at a daily temporal resolution. Low river flow is described as the percentile of daily streamflow that is exceeded 90% of the time. It is determined separately for each GCM/HM combination and the warming scenarios. The results show that the low flow change signal amplifies with increasing warming levels. Low flows decrease in the Mediterranean, while they increase in the Alpine and Northern regions. In the Mediterranean, the level of warming amplifies the signal from -12% under 1.5 K, compared to the baseline period 1971-2000, to -35% under a global warming of 3 K, largely due to the projected decreases in annual precipitation. In contrast, the signal is amplified from +22% (1.5 K) to +45% (3 K) in the Alpine region due to changes in snow accumulation. The changes in low flows are significant for regions with relatively large change signals and under higher levels of warming. However, it is not possible to distinguish climate induced differences in low flows between 1.5 and 2 K warming because of (1) the large inter-annual variability which prevents distinguishing statistical estimates of period-averaged changes for a given GCM-HM combination, and (2) the uncertainty in the multi-model ensemble expressed by the signal-to-noise ratio. The contribution by the GCMs to the uncertainty in the model results is generally higher than the one by the HMs. However, the uncertainty due to HMs cannot be neglected. In the Alpine and Northern region as well as the Mediterranean, the uncertainty contribution by the HMs is partly higher than those by the GCMs due to different representations of processes such as snow, soil moisture and evapotranspiration. Based on the analysis results, it is recommended (1) to use multiple HMs in climate impact studies and (2) to embrace uncertainty information on the multi-model ensemble as well as it's single members in the adaptation process.

## 1 Introduction

Hydrological drought is a slowly developing natural phenomenon than can occur anywhere, independently of the hydro-climatic regime (Van Loon, 2015). It is expressed as a deficiency in river discharge compared to the expected normal and

is mainly caused by lower than average precipitation and soil moisture or strong increases in evapotranspiration. In addition to natural causes, human water use and reservoirs can significantly alter the drought signal in many places (Wanders and Wada, 2015). Droughts are rare events and can propagate from meteorological over soil moisture to hydrological droughts, finally resulting in socio-economic drought (Van Loon, 2015). Hydrological droughts affect the environment and cause damage to society and the economy. van Vliet et al. (2016) showed reduced potentials for thermoelectric power and hydropower generation

under hydrological drought worldwide. In Europe, the 2003 drought and heatwave resulted in nearly -6.6% in hydropower and -4.7% in thermoelectric power generation. The total loss of the 2003 severe drought event was estimated to be 8.7 billion Euro in Central and Southern Europe (EC, 2007). More recently, the 2015 drought event (Laaha et al., 2017; Van Lanen et al., 2016; Zink et al., 2016) in Central Europe also caused significant socio-economic and environmental problems. Economic losses due to droughts almost doubled between 1976-1990 and the 1991-2006 period to approximately 6.2 billion Euro per year. Social

and environmental costs are often not considered (EC, 2007). A collection of hydrological drought impacts for Europe can be found in the European drought impact inventory (Stahl et al., 2016) sorted by impact categories, e.g., freshwater aquaculture and fisheries, energy and industry, waterborne transportation, public water supply or freshwater ecosystems. Furthermore, water quality is directly influenced by hydrological drought, e.g., in lowering the availability of the diluting medium water resulting in increasing pollutants concentrations.

Climate change is expected to alter the hydrological cycle throughout Europe. Temperature projections show significant warming for all emission scenarios over Europe. Southern Europe is the hotspot with strongest projected warming in summer, Northern Europe in winter time (Kovats et al., 2014). Jacob et al. (2014) projected mean annual precipitation to decrease under RCP4.5 mainly in the Iberian Peninsula and Greece until the end of the century. It is expected that large areas from the UK over France and Italy to the Balkan states will experience almost no annual precipitation changes, whereas Central Europe and

Northern Europe face precipitation increases. Under RCP8.5, the signal intensifies with an increase in large parts of Central Europe and Northern Europe of up to approximately 25% and a decrease in Southern Europe. Meteorological droughts are projected to occur more frequently in the Mediterranean and to become less frequent in Scandinavia, with an intensification of the signal with increased warming levels (Stagge et al., 2015).

In the Paris Agreement of 2015, the Conference of the Parties of the United Nations Framework Convention on Climate Change

emphasised "holding the increase in the global average temperature to well below 2°C above pre-industrial levels and pursuing efforts to limit the temperature increase to 1.5°C above pre-industrial levels" (UNFCCC, 2015) and invited the Intergovernmental Panel on Climate Change (IPCC) to prepare a special report on the impacts of a global warming of 1.5°C in 2018. Notably, based on the estimated emissions over the past decades, it remains unclear if a limitation of global warming to two

degrees or even three degrees can be achieved (Peters et al., 2012). Most climate impact studies in the past have focused on future time periods, e.g., changes until 2071-2100 under different emission scenarios or representative concentration pathways (RCPs). Mitchell et al. (2016) argue that these studies are hardly usable for determining differences between warming levels, partly because of the large internal range of warming within the RCPs. Collins et al. (2013) reported the likely range of global warming for 2081-2100 relative to 1986-2005 for the CMIP5 models with 0.3 K to 1.7 K under RCP2.6, 1.4 K to 3.1 K under RCP6.0, and 2.6 K to 4.8 K under RCP8.5.

In recent literature, several studies have investigated climate impacts on low flows and hydrological droughts in Europe, focusing on differences between historical and future time periods (e.g., van Vliet et al., 2015; Wanders et al., 2015; Forzieri et al., 2014; Schneider et al., 2013). Recent assessment studies have changed their focus more towards analysing warming levels, covering the 2 degree goal (Roudier et al., 2016), comparing impacts between different levels of warming in selected river basins (Gosling et al., 2017), or focusing on runoff rather than streamflow (Donnelly et al., 2017). This study investigates projected changes in low streamflow, defined as Q90, representing daily streamflow exceeding 90% of the time, which has the potential to impact hydrological drought. Hydrological drought is associated with shortfalls on surface or subsurface water availability which can occur in low streamflow, groundwater, or reservoir levels. Changes in low flows analysed in this study can, but not always, result in drought. Exceptions are e.g., riverine based transportation, where streamflow values below a threshold level are defined as hydrological drought.

Whilst the climate and hydrological models in the available studies vary significantly as well as the formulation of low flow indices, similar patterns could be found. Decreasing river low flows are projected in Southern Europe and increasing low flows in Northern Europe. Nevertheless, there are limited studies reporting on changes in low flow conditions across Europe using an ensemble of GCM/HM simulations at high spatial resolution and for different warming levels. We fill this gap by analysing the changes in low flow conditions based on a large ensemble of hydrological simulations conducted at a high spatial resolution (5km) over Europe for different warming levels.

Specifically, we provide a comprehensive impact and uncertainty assessment for hydrological low flows across Europe under a global warming of 1.5, 2, and 3 K. The study is based on a multi-member ensemble of high resolution simulations ($5 \times 5$ km$^2$) from the EDgE project (http://edge.climate.copernicus.eu, End-to-end Demonstrator for improved decision making in the water sector in Europe) which has been enlarged to 45 ensemble simulations consisting of three hydrological models (HMs) driven by five General Circulation Models (GCMs) under three RCPs. A consistent setup is achieved using identical meteorological input and land surface data to establish the three HMs. To investigate the usability of the simulation results, information on the robustness and uncertainty of projected changes as well as GCMs and HMs contributions to the overall uncertainty are discussed. The research questions aim to close a knowledge gap with respect to impacts of different levels of climate warming are as follows:

1. What is the magnitude and robustness of change in low flows in Europe under a global warming of 1.5, 2, and 3 K?

2. Is there a significant difference in projected changes of low flows between the three global warming levels?

3. How much do the GCMs and HMs contribute to the overall uncertainty for the particular warming levels?

## 2    Material and Methods

The study presented herein uses a consistent set of 45 high-resolution hydrological simulations based on five GCMs under three RCPs driving three HMs across Europe at a 5 km spatial resolution. The aim is to provide a consistent framework using a compatible set of standardised forcings and initial conditions for the impact models to investigate low flow changes under different levels of warming. This multi-model ensemble has recently being used to analyse projected changes in river floods and high flows in Europe by Thober et al. (2017).

### 2.1    Climate and hydrologic models

Five CMIP5 General Circulation Models (GCMs: HadGEM2-ES, IPSL-CM5A-LR, MIROC-ESM-CHEM, GFDL-ESM2 and NorESM1-M) provided temperature and precipitation data to drive three hydrological models (HMs). Data for the time period 1950 to 2099 at a daily time step is available under three Representative Concentration Pathways (RCPs: 2.6, 6.0, and 8.5) from the ISI-MIP project (Warszawski et al., 2014, data available under doi:10.5880/PIK.2016.001). A trend-preserving bias-correction is applied to GCM data by Hempel et al. (2013). GCM data at a horizontal resolution of $0.5°$ is hardly applicable to describe land surface processes on catchment scales in Europe. Therefore, this data has been disaggregated to $5 \times 5$ km$^2$ using External Drift Kriging (EDK) and the elevation as external drift within the EDgE project. This interpolation technique accounts for altitude effects in temperature and precipitation and is widely applied in hydrological simulations (Zink et al., 2017). EDK adds sub-grid variability to the GCM fields, reflecting e.g., the altitude dependency of temperature. Methods such as EDK generally perform better in interpolating continuous meteorological variables compared to discontinuous variables such as precipitation. It is worth noting that the long-term trends are preserved using this interpolation technique. The variogram for EDK is estimated using the original E-OBS station data.

This meteorological data set at a spatial resolution of $5 \times 5$ km$^2$ is then used to force the three HMs: mHM, Noah-MP, and PCR-GLOBWB. Within the EDgE project, the HMs have been consistently set-up using the same land surface datasets (terrain, land cover, soil maps and geological information). Furthermore, a consistent external river flow routing scheme has been applied to outputs of all HMs based on the multiscale Routing Model that has been developed originally for mHM (Samaniego et al., 2010). Ultimately, the differences in the hydrological simulations result from different process representations and parameterisations of the surface and subsurface in the HMs.

The HMs used in this study are grid-based distributed models grounded on numerical approximations of dominant hydrologic processes. The mesoscale Hydrological Model (mHM, Samaniego et al., 2017b) has originally been developed in Central Europe and it uses the multiscale parameterisation technique, MPR (Samaniego et al., 2010; Kumar et al., 2013); that allows the model applicability at different spatial resolutions ($1 \times 1$ km$^2$ to $50 \times 50$ km$^2$) and multiple locations without much of a calibration effort. The Noah-MP model was originally developed as land surface component of the 5th generation mesoscale model MM5 to enable climate predictions with physically based ensembles and represents both the terrestrial water and energy cycle (Niu et al., 2011). The PCRaster global water balance model (PCR-GLOBWB) was developed to represent the terrestrial

water cycle with a special focus on groundwater and modelling water resources under water stress (Van Beek and Bierkens, 2008; Wanders and Wada, 2015).

The three HMs used in this study are calibrated in nine near-natural European focus basins located in Spain, Norway and UK, which are selected based on the consultation with the user groups within the EDgE project. Besides these, we also include three more central EU catchments (located in France and Germany) to represent diversity in hydro-climatic regimes. All HMs parameters are calibrated such that the model simulations represent a range of hydrologic regimes, rather than tailored to any specific characteristics. This is done in a consistent manner so that the model simulations can be used for a range of indicators (including high, low, and average flows) within the EDgE project, resulting in slightly lower performances for low flows. We note that HMs could be calibrated to specific parts of the flow duration curve (FDC), however, this is not done in this study to avoid too specific tuning of the model simulations to those unique conditions and thereby losing valuable information on the entire FDC. In the current simulations human water management was not taken into account, since some models lack the ability to include these processes. Human water management can however have a significant impact on the low flow conditions, due to abstraction of additional water in drought condition or changes in reservoir management. As a result constraining the model to any specific low flow characteristic can result in a biased simulation. Also due to the similar reason we may expect a relatively lower model skill in matching observed low flow characteristics. The HMs are calibrated using observation-based E-OBS data (V12.0, Haylock et al., 2008) and automatic calibration schemes are employed for mHM (Rakovec et al., 2016) and PCR-GLOBWB. Noah-MP has been calibrated manually adjusting the parameter for evaporation surface resistance based on the analysis by Cuntz et al. (2016).

Temperature and precipitation data from GCMs with coarse resolution have different statistical properties than interpolated observational datasets. To investigate if the observation-based calibration of the HMs is applicable to the disaggregated GCM data, model outputs are evaluated against 357 gauging stations using the GCM forcing during the historic period 1966-1995 (Fig. 1). The stations and time period are selected to ensure the largest possible, complete dataset over 30 years. Their median basin area is 1680 km$^2$. The analysis focuses on matching the median of the 30 years annual percentile for low flows (Q90). The indicator for low flows is used herein for the impact assessment studies as detailed described in section 2.3.

The evaluation results show overall an overestimation of observed Q90 by all HMs and GCMs. (Figure 1, lower left). This overestimation in the ensemble average is mainly the result of the overestimation by the HMs PCR-GLOBWB and Noah-MP simulations, while the mHM runs show only a slight overestimation and result in closest correspondence to the observed values. Nevertheless, it cannot be concluded that mHM performs best due to the neglection of human activities in many basins (abstraction as well as e.g., ensuring minimum ecological flow). Well-calibrated HMs do not necessarily mean that future simulated discharge under a changed climate can be reproduced satisfactorily (Vaze et al., 2010). Furthermore, the selection of HMs may have a larger effect than the calibration of parameters in hydrological climate impact studies (Mendoza et al., 2015). The spatial pattern of the relative bias for the multi-model ensemble average is shown in Fig. 1 (lower right). It is important to assert that this spatial pattern differs significantly between the HMs while the climate change signal for low flow projections in this study (see section 3) is remarkably similar across all three HMs.

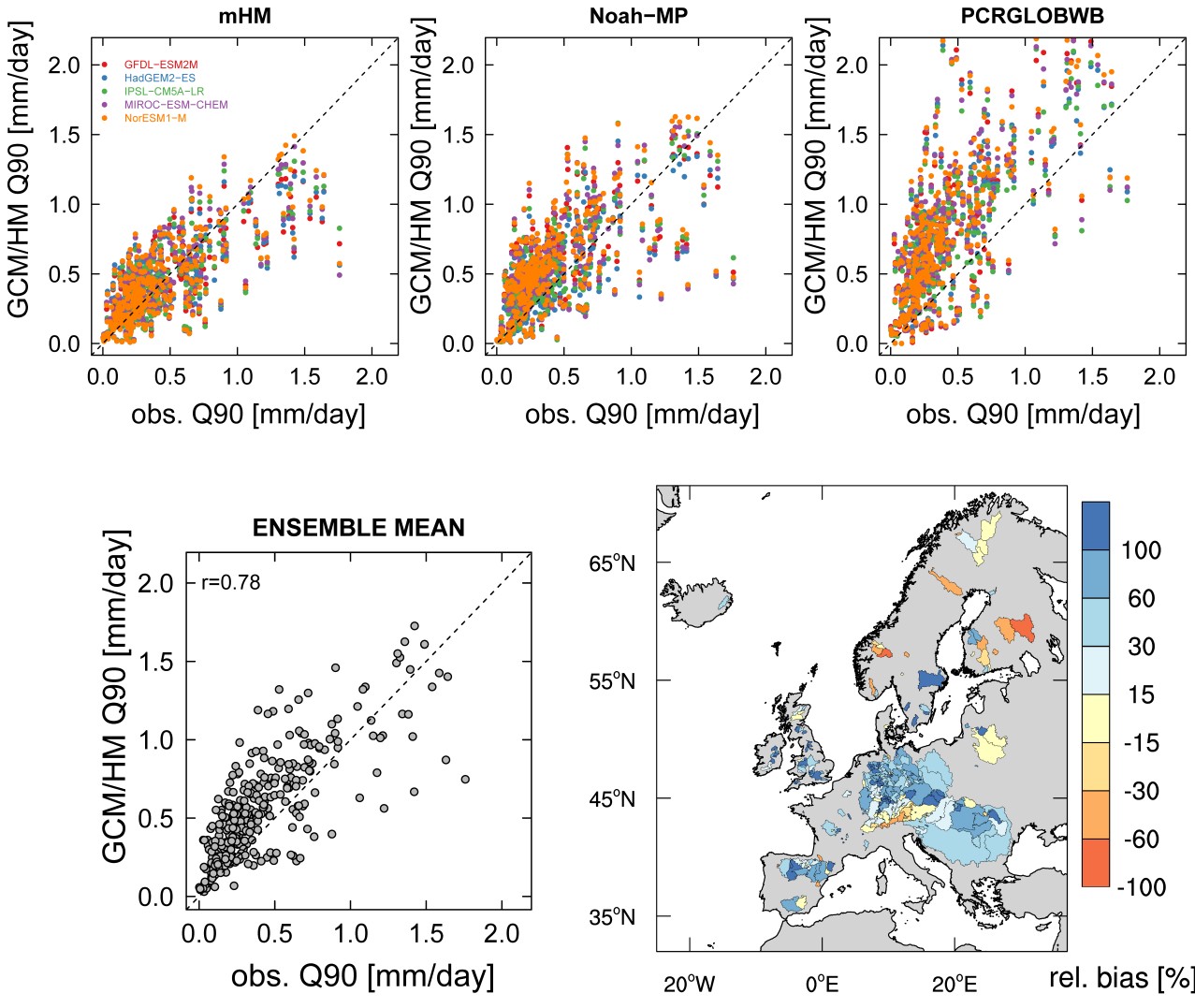

**Figure 1.** Scatter plot between observed low flow and GCM-HM simulated low flow (Q90) over 357 gauges across Europe. Simulated values correspond to the median of the annual estimates calculated for the historical time-period 1966-1995. The colours of the dots denote the five GCMs used to drive the hydrologic models mHM (left column), Noah-MP (middle column) and PCR-GLOBWB (right column). The location of the basins and the spatial pattern of the relative bias is shown on the lower right.

## 2.2 Determination of 1.5, 2, and 3-K time periods

The five CMIP5 GCMs used in this study have different sensitivities to climate forcing. The development of annual global temperature varies significantly over time between the models and RCPs. Therefore, the time period with a mean global warming of 1.5, 2, and 3 K with respect to pre-industrial condition also varies between the GCM simulations. Here, a time

**Table 1.** Determination of 1.5, 2, and 3 K time periods for different GCM/RCP combinations. A time sampling approach was used comparing 30-year running means to the period 1971-2000 with an assumed warming of 0.46 K to pre-industrial conditions.

| Warming level | RCP | GFDL-ESM2M | HadGEM2-ES | IPSL-CM5A-LR | MIROC-ESM-CHEM | NorESM1-M |
|---|---|---|---|---|---|---|
| | 2.6 | - | 2007-2036 | 2008-2037 | 2006-2035 | 2047-2076 |
| 1.5 K | 6.0 | 2040-2069 | 2011-2040 | 2009-2038 | 2012-2041 | 2031-2060 |
| | 8.5 | 2021-2050 | 2004-2033 | 2006-2035 | 2006-2035 | 2016-2045 |
| | 2.6 | - | 2029-2058 | 2060-2089 | 2023-2052 | - |
| 2 K | 6.0 | 2060-2089 | 2026-2055 | 2028-2057 | 2028-2057 | 2054-2083 |
| | 8.5 | 2038-2067 | 2016-2045 | 2018-2047 | 2017-2046 | 2031-2060 |
| | 2.6 | - | - | - | - | - |
| 3 K | 6.0 | - | 2056-2085 | 2066-2095 | 2055-2084 | - |
| | 8.5 | 2067-2096 | 2035-2064 | 2038-2067 | 2037-2066 | 2057-2086 |

sampling method is used to determine the time-period for different levels of global warming (James et al., 2017). This approach has been used to investigate climate impacts over Europe for a global warming of 2 K (Giannakopoulos et al., 2009; Vautard et al., 2014) and for global differential impacts between a warming of 1.5 K and 2 K (Schleussner et al., 2016). 30-year running mean global temperatures are compared to those of the 1971-2000 period in the GCM simulations. The latter period
corresponds to a global warming of 0.46 K (average value from three estimations with a spread between 0.437 K and 0.477 K) with respect to pre-industrial condition (Vautard et al., 2014). The first 30-year period with a global warming crossing one of the three warming levels (1.5, 2, 3 K) is then determined for each of the 15 GCM/RCP combinations. The identified 30-year time-period for the corresponding GCM/RCP combination is shown in Table 1. It is worth noting that for some of the combinations, we could not identify any 30-year period for a selected warming level. For example, none of the GCM simulations crossed the
3 K warming level under the RCP2.6 over the entire simulation period up to 2099.

Available methods for identifying regional climate responses to global warming targets face advantages and disadvantages (James et al., 2017). Limitations in the time sampling method occur in the direct comparison between different warming levels because the number of ensemble members varies. Available simulations reduce from 14 under 1.5 K warming over 13 under 2 K to 8 simulations under a global warming of 3 K. Furthermore, the annual temperature within future 30-year periods may be
pathway dependent, e.g., a rapid or slower warming. This may influence the results in climate impact simulations. Nevertheless, the time sampling method is advantageous creating a large ensemble of simulations, which is essential to determine differences between warming levels (Mitchell et al., 2016).

### 2.3 Low flow indicator used, uncertainty metrics, and spatial aggregation of results

The impact of climate change is quantified for low flows. Commonly, the 70th to 90th percentile of exceedance is used to define
hydrological droughts for rivers with perennial type streamflow (Fleig et al., 2006). Within the framework of the EDgE project,

the co-production with stakeholders from the water sector in Norway, Spain and the UK resulted in Q90 (daily flows exceeded 90% of the time) as low flow index. The Q90 is estimated for each calendar year over a given 30-year period, and the median of Q90 is subsequently calculated from the respective 30 samples as a final indicator. We recognise that the use of a calendar year may influence our results in snow-influenced catchments where the low flow period may span over two consecutive years.

To assess possible consequences, we compared the annual results against simulations for the winter half year and found only minor changes in overall results, especially in snow dominated regions. Further seasonal assessment is not performed in this study. We use the period 1971-2000 as a reference for the estimation of climate impacts and the relative changes in Q90 is estimated with respect to this reference period for different warming levels.

The non-parametric Wilcoxon rank-sum test is applied to account for the robustness of the results. The null hypothesis of equal
means between the climate periods per GCM-HM simulation is tested at 5% significance, which has been applied in Gosling et al. (2017) among others. Based on the ensemble of Wilcoxon rank-sum tests, the robustness is estimated following the IPCC AR4 procedure presented in Solomon et al. (2007). Robustness is computed as the percentage of projections showing a significant change. Important thresholds are less than 33% for unlikely and greater than 66% for likely changes, representing the percentage of ensemble simulations showing a significant change. Significance here does not account for the sign or magnitude
of change.

The signal to noise ratio (SNR) is commonly used to quantify the uncertainty in hydrological extremes studies (Prudhomme et al., 2014; Hall et al., 2014; Giuntoli et al., 2015). Here, the SNR is computed as the median divided by the inter-quantile range (i.e., the difference between the 25th and 75th percentile). It has been acknowledged in recent literature that both GCMs and HMs contribute to the uncertainty in projected changes (Gosling et al., 2017; Donnelly et al., 2017; Hattermann et al.,
2017). In this study, the sequential sampling approach of Samaniego et al. (2017a), following Schewe et al. (2014), is applied. In this approach, the uncertainty due to GCM is estimated by first fixing a HM and then calculate the range (max-min) of Q90 changes corresponding to five GCMs outputs. Repeat the previous step for all other remaining HMs. Finally, estimate the average of ensemble ranges that would then represent the uncertainty due to GCMs. Likewise, the same steps could be repeated by fixing the GCM and calculating the range statistics over the HMs to represent the uncertainty component due to HMs. We
use the bootstrap technique to account for different sample size of GCM and HMs; and perform the sequential uncertainty assessment with three GCMs and HMs outputs over the 1000 realizations.

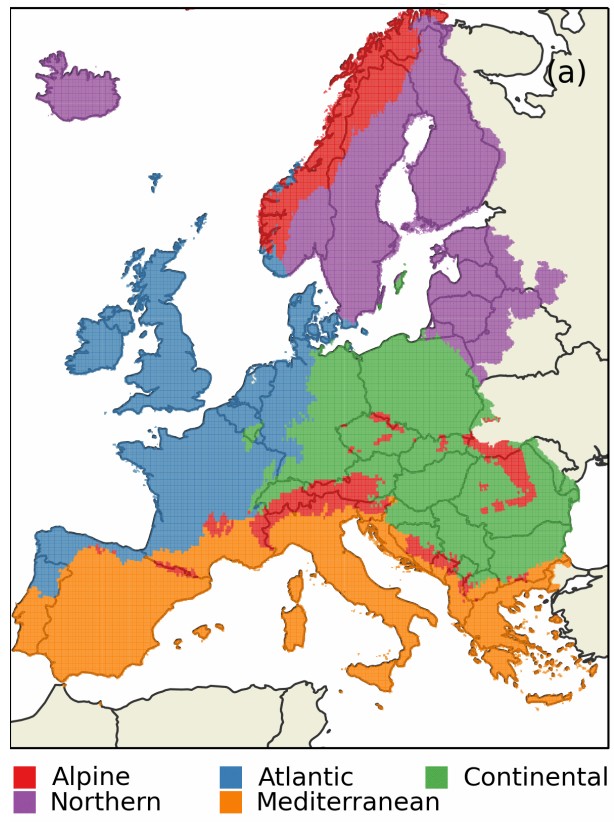

**Figure 2.** European macro-regions used in the IPCC AR5 (Kovats et al., 2014) based on an environmental stratification after Metzger et al. (2005) (Source: own graphics based on GIS data provided by Marc J. Metzger, University of Edinburgh. The data is remapped to the 5 km grid used in this study).

To account for regional differences in climate impacts, the results of our analyses are displayed over Europe and additionally aggregated for five different regions (Fig. 2). These macro-scale regions have been used in the latest IPCC WGII report for Europe (Kovats et al., 2014) and were originally identified based on the environmental stratification presented in Metzger et al. (2005), using a principal component analysis accounting for 20 different environmental variables. Furthermore, the low flow impact assessment carried out here is limited to river basins with upstream areas greater than 1000 km$^2$. Smaller (and headwater) basins are not considered here as to limit the delineation errors of river network in the runoff routing scheme (see e.g., Fig. 3 for the resulting river network).

**Table 2.** Relative changes [%] in streamflow Q90 between the past (1971-2000) and different warming levels averaged over IPCC AR4 Europe regions shown in Fig. 2.

| Warming level | Absolute warming | Alpine | Atlantic | Continental | Northern | Mediterranean |
|---|---|---|---|---|---|---|
| 1.5 K | 1.04 K | 22.2 | -7.3 | -4.1 | 8.4 | -12.0 |
| 2 K | 1.54 K | 29.6 | -10.0 | -4.5 | 15.9 | -16.3 |
| 3 K | 2.54 K | 44.8 | -21.6 | -19.1 | 24.1 | -35.1 |

## 3   Results and Discussion

### 3.1   Changes in low flows under different levels of warming compared to 1971-2000

The change signal in low flows gets stronger with increased levels of warming in most parts of Europe (Fig. 3, left row). An amplification in decreasing low flows can be identified in the Iberian Peninsula, the south-western part of France, and southeast Europe including Greece and the Balkan states. On the contrary, large parts of the Alps and Scandinavia face an intensification of increasing low flow signal with higher levels of warming. The region from Germany over Poland to the Baltic countries shows generally very small changes, and the sign of change in low flows alters with increased warming. Under a global warming of 1.5 K, the mean change in streamflow Q90 over Europe is approximately zero (Fig. 3, upper left), but with large spatial differences between the IPCC AR5 Europe regions and with different directions of change. The regional low flow statistics are based on the average of all the grid cells per region.

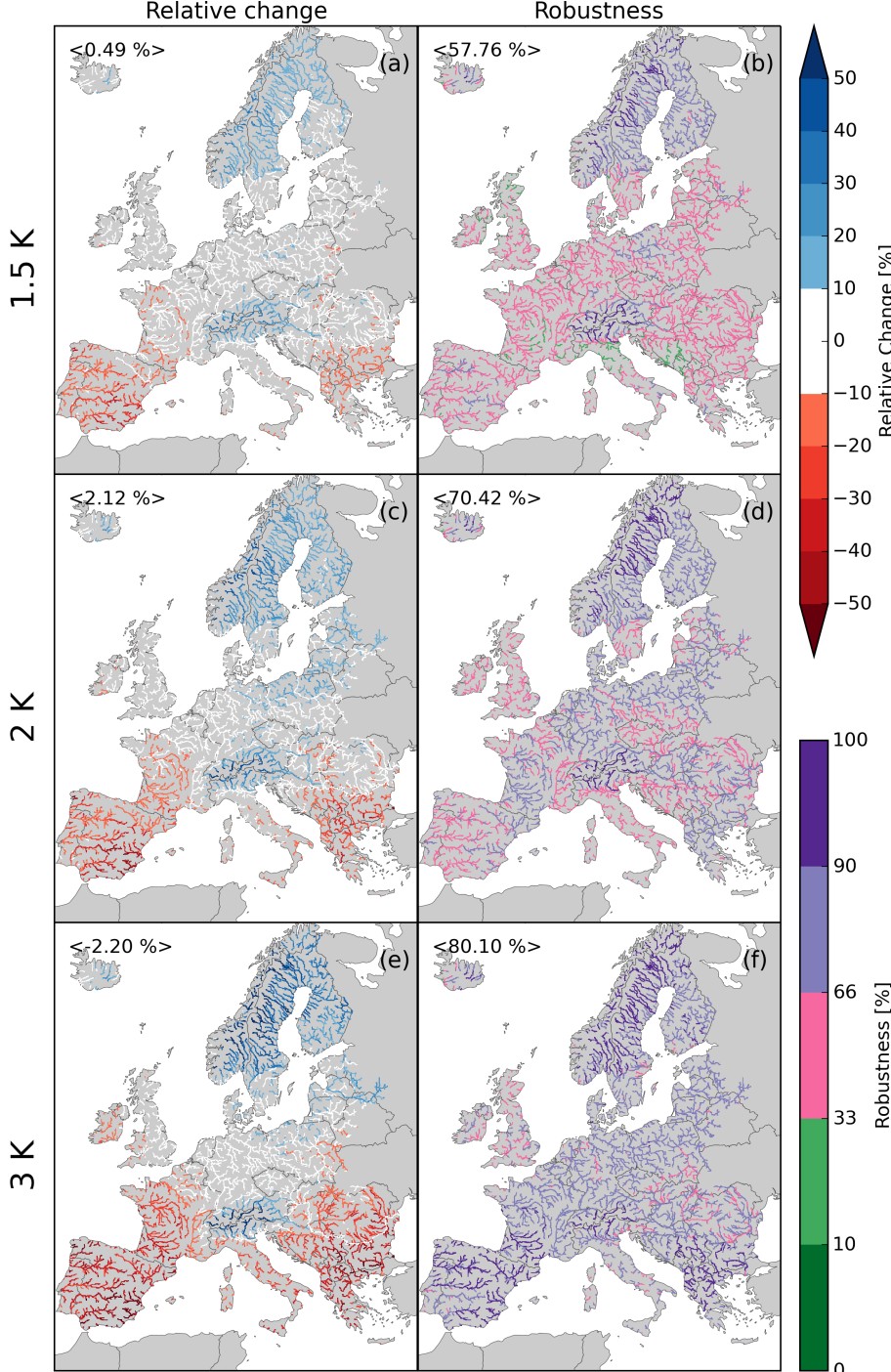

**Figure 3.** Change in multi-model ensemble mean low flow [%] under different warming levels compared to the 1971-2000 baseline (left) and robustness (right). The latter is expressed by the percentage of simulations based on a Wilcoxon rank sum test with 5% significance level. An agreement of more than 66% in the ensemble is classified as "likely" change. The values given in the upper left of the subplots is the continental average along the river network for all grid cells with a contributing area greater than 1000 km$^2$.

Approximately half of the rivers in Europe show decreases in low flows under 1.5 K warming, with an hotspot in the Iberian Peninsula region and the strongest decrease in the Mediterranean [-12% over the whole area] and the Atlantic region [-7%] (Tab. 2). On the contrary, increases in low flow are expected in the Alpine [+22%] and Northern areas [+8%]. This occurs mainly due to changes in snow accumulation and melt, and consequently results in higher winter low flows. The Continental area shows overall the smallest changes with both positive and negative values, but less than 10% even under a global warming of 2 K.

More regions in Europe show significant changes in low flow with an increased level of warming (Fig. 3, left row). Robustness is expressed as the percentage of simulations passing the Wilcoxon rank sum test at 5%. Under a warming level of 1.5 K, approximately 57% of the ensemble simulations show significant changes. Highest values are found in snow-dominated regions (e.g., Alpine and Northern region). Under a warming level of 2 K, the percentage of ensemble simulations with significant changes increases to approximately 70%, being distributed equally over Europe, and this number increases to 80% for a warming level of 3 K. Under a global warming of 3 K, the agreement among the ensemble simulations increases to overall 80%. The strongest regional change is found in the Mediterranean, with likely changes across 31% of the river basins under a warming level of 1.5 K, 64% under 2 K and 90% under 3 K, respectively. The significance is highest in regions with strong (positive and negative) change signals. Nevertheless, there are exceptions, e.g., under 2 K warming the signal for the Mediterranean might be stronger, but it is less robust than that for the Atlantic.

The results presented here confirm those found in earlier studies for low flow and hydrological drought projections across Europe. Forzieri et al. (2014), for example, gave an overview on projected changes in average 7-day minimum flows until the end of the century under the SRES A1B scenario. A single HM was selected for the analysis in that study which was then driven by 12 regional climate model (RCM) precipitation and temperature dataset. The analysis showed that streamflow droughts become more severe and persistent in Southern Europe, while droughts decrease in northern and northeastern parts of Europe. Wanders et al. (2015) found similar patterns over Europe using 5 GCMs and a single HM, with a clear influence of decreasing snow accumulation in Northern Europe and an increase in drought impacts in the Mediterranean. Recently, Gosling et al. (2017) investigated changes in hydrologic droughts under a global warming of 1, 2 and 3 K over large river catchments (greater than 50000 km$^2$) including two European basins - the Central European Rhine and Mediterranean Tagus River. They used Q95 as a low flow indicator, based on the same 5 GCMs applied in our study with an ensemble of global as well as catchment hydrological models. Nevertheless, the results from both studies are comparable under a global warming of 2 K and 3 K, with projected decrease in low flows in the Rhine and Tagus River. Low flow (Q90) in this study under a warming level of 2 K is almost unchanged in the Rhine, and up to -11% under 3 K. The more pronounced low flow decrease is found in the Tagus River showing -16% under 2 K and -33% under a global warming of 3 K. The GCMs used in van Vliet et al. (2015) are also identical to those used in this study. However, the HMs E-HYPE (Donnelly et al., 2016) and VIC (Cherkauer et al., 2003) were used to simulate the changes in Q90 for RCP2.6 and RCP8.5 for the 2050s and 2080s. Overall, the spatial pattern of changes in flow indicator fits to our results quite well, likewise the amplification of the signal over time until the end of the century was also found in both studies. The strongest reductions in low flows are exhibited in Southern Europe and related to decreasing annual precipitation. The spatial pattern under a global warming of 2 K compares well with those reported by Roudier et al.

(2016) for low flows with a 10-year return period. Notably, the underlying model ensemble consists of 11 bias-corrected RCMs and two hydrologic models, which are different from those used in this study. They found a 15% reduction in low flows for the Mediterranean, which is very similar to the 16% reduction found in this study. Although the results on the climate induced change in low flows presented herein are generally comparable to other studies, we provide new spatially explicit information

on low flows under different levels of warming over Europe.

Our study shows contrasting results for the Mediterranean region compared to Donnelly et al. (2017) under different levels of warming. At a global warming of 3 K, large decreases up to -35% and high robustness (very likely) are observed here, whereas no projected changes in absolute grid-specific runoff values with little robustness was reported by Donnelly et al. (2017). These differences can be explained through methodological choices on low flow indices used between the two studies. The relative

changes in the routed river low flow quantified here is more informative for water resources assessments compared to the absolute changes of grid-specific runoff. This holds especially true in drier regions, which are characterized by very small Q90 runoff values. From a practitioner point of view, our study highlights the need for adaptation to climate induced low flows in these regions, which would not be concluded based on the metrics reported in Donnelly et al. (2017).

It is observed that the changes in river low flows can be explained to a large extent by the median change in annual precipitation

over all levels of global warming (Fig. 4). To investigate the influence of precipitation on low flows, we compare the relative change of Q90 discharge to the changes in the annual total precipitation over the 30-years for different levels of warming. The Mediterranean region shows the strongest decrease in precipitation and low flows among all warming levels. The correlation coefficient between changes in annual precipitation and Q90 increases from 0.45 under 1 K to 0.62 under 3 K level of warming. Notably, the increased spread in the median changes of annual total precipitation and simulated river low flows under the

warming level of 3 K contribute to higher correlation compared to other warming levels. Furthermore, we observe a relatively stronger correspondence between changes in annual total precipitation and low flow indicator in river basins characterized by projected decrease in low flows. The $r^2$ value rises from 0.61 to 0.77 with an increase in a global warming level from 1.5 to 3 K; compared to an increase of 0.45 to 0.65 for the same warming levels in river basins showing projected increase in low flows. Overall, the Continental and Atlantic regions show the smallest changes in precipitation and low flows. In the Northern region,

the projected increases in changes of both variables are highest. In this region, the relationship between precipitation and low flows is the weakest as exemplified by the low $r^2$ values for the positive precipitation changes. This can be explained due to the increasing influence of snow processes, accumulation as well as snow melt. This holds also true for catchments greater than 1000 km$^2$ in Alpine regions (not displayed here).

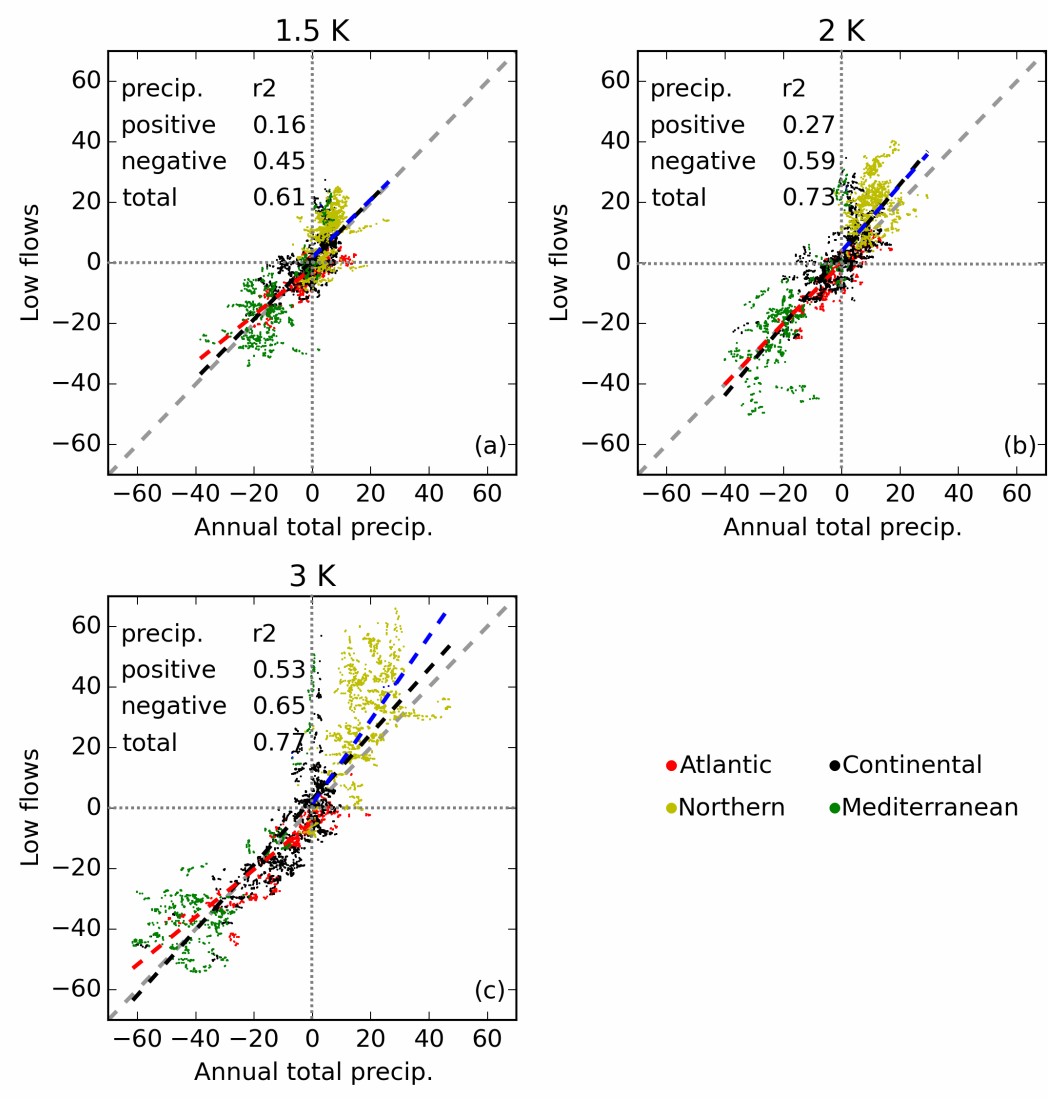

**Figure 4.** Relationship between the median changes in the annual total precipitation and simulated river low flows (Q90) under a global warming of 1.5 K (a), 2 K (b) and 3 K (c). Anomalous to other results shown in this study, only river grid cells from basins greater than 10000 km$^2$ are shown for clarity in the figure. Results are similar to those including river grid cells with contributing areas greater than 1000 km$^2$. Linear regression lines are shown for positive values (blue dashed), negative values (red dashed) and all data points (black dashed). The Alpine with overall smaller catchment sizes is not included, but shows a similar behaviour to the basins in the Northern region. All changes are expressed as multi-model ensemble mean changes (GCM/HM combinations for low flows and GCMs for annual precipitation).

**Table 3.** Relative changes averaged over regions [%] in multi-model ensemble mean low flow indicator (Q90) between different levels of global change.

| Warming level | Absolute warming | Alpine | Atlantic | Continental | Northern | Mediterranean |
|---|---|---|---|---|---|---|
| 1.5 K → 2 K | 0.5 K | 8.6 | -1.1 | -0.3 | 10.7 | -6.6 |
| 2 K → 3 K | 1.0 K | 17.0 | -9.0 | -12.3 | 13.5 | -16.0 |
| 1.5 K → 3 K | 1.5 K | 23.9 | -12.9 | -12.2 | 22.6 | -24.0 |

Under a warming level of 3 K, we identified a larger spread between total annual precipitation and low flows. In the Northern area, this can be explained due to higher temperatures which could then lead to less snow accumulation and increased winter low flows. In contrast, higher temperatures combined with lower than average annual precipitation in the Mediterranean result in higher evapotranspiration and decreased low flows. Our results agree with other studies reporting about the general relation-
ship between precipitation and low flow changes (e.g., Forzieri et al., 2014; van Vliet et al., 2015; Gosling et al., 2017), even though relating precipitation and low flows in different ways. In the following section, the differences between policy relevant levels of warming are examined.

## 3.2 Differences in low flows between different future levels of warming

One of the objectives of this study is to analyse differences in the change signal and the sensitivity of the low flow changes
to different levels of global warming. This provides additional information compared to the results presented above. Both of these results, in combination, are important for the discussion on mitigation targets and for adaptation planning in accordance with the Paris agreement (UNFCCC, 2015). With increased levels of global warming from 1.5 to 2 K, 2 to 3 K and 1.5 to 3 K, an amplification of the change signal in low flow is expected over a large part of Europe (Fig. 5, panels a, c, and e). This holds especially true in regions with relatively big positive and negative changes in low flows. The overall robustness of the low flow
changes in Europe increases with increasing temperature differences between the global warming levels (Fig. 5, panels b, d, and f).

The changes in streamflow Q90 between 1.5 K and 2 K warming are generally small with few rivers exhibiting changes larger than 10% in magnitude. The pattern is similar to the one shown in Fig. 3, which highlights that the sign of change is conserved in areas with relatively large changes (more than ±10%), even under the small warming of only 0.5 K. These results, however,
are not robust. None of the rivers show likely changes, meaning that less than 66% of the ensemble simulations are significant at the river grid cell level. Moreover, most parts of Europe show changes marked as unlikely with a total agreement of only 15% over Europe and all simulations. The regional changes in low flows between the two warming levels are also small (see Tab. 3). The Atlantic and Continental area show almost an unchanged situation. The Northern region exhibits the largest increase with 11%, and the Mediterranean faces -7% decrease in low flows averaged over the considered stratified region.

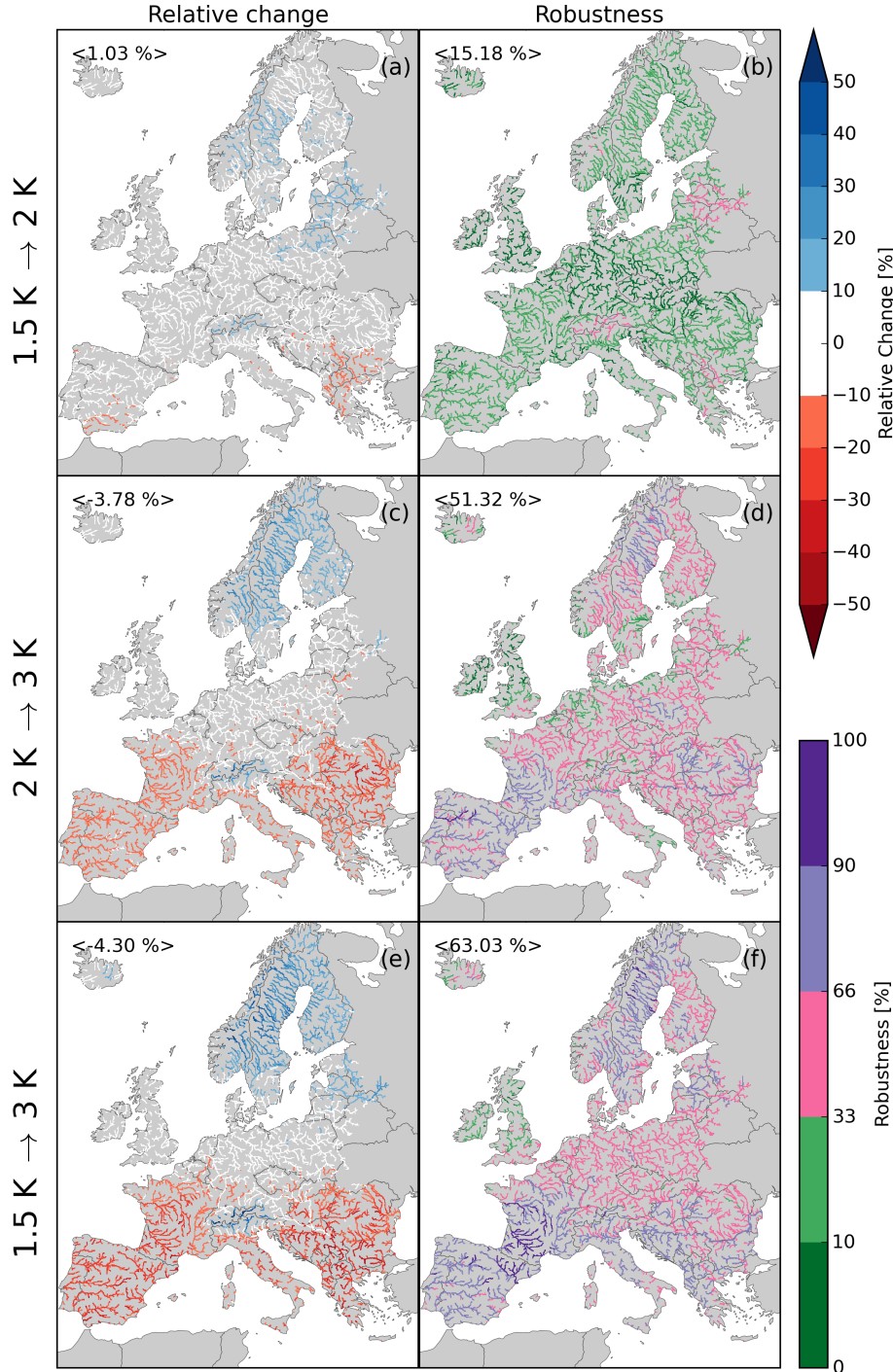

**Figure 5.** Relative change [%] in multi-model ensemble median Q90 between different levels of warming (left) and robustness of the signal between those (right). The latter is expressed by the percentage of simulations based on a Wilcoxon rank sum test with 5% significance level. The values given in the upper left of the subplots is the continental average along the river network for all grid cells with a contributing area greater than 1000 km$^2$.

The robustness results presented in Fig. 3 (panels b, d, f) alone do not allow for determining warming level thresholds of change in low flow indicator. Therefore, we included the robustness of the change between the warming levels in this section. Combining the information in Fig. 3 (b,d) with Fig. 5 (b), we see robust changes between the past time period and a 2 K warmer world. The information of non-significant differences between 1.5 K and 2 K warming allows for the conclusion that the majority of change already happens before reaching a warming level of 1.5 K. Limiting climate change to a warming level of 1.5 K in comparison to 2 K has only a limited effect on low flows. These results point out that an even lower mitigation goal would be needed for regions where substantial negative impacts occur.

Low flow changes between 2 K and 3 K warming (Fig. 5, panels c and d) are more pronounced with large parts of the Central Alps and Scandinavia showing an increase of more than 10% in low flows. On the contrary, most regions on the Iberian Peninsula, France, Italy, the Balkan states and Greece face a decrease of more than 10% low flow. The strongest increase is projected for the Alpine region (+17%), and the strongest decrease for the Mediterranean (-16%). Overall, half of the simulations show robust changes over Europe with large regional differences. Likely changes are found in the southwest of Europe, northern Norway and the Balkan states. It is worth emphasising that the differences between a global warming of 2 K and 3 K in low flows are substantial. These changes are on top of those projected between 1971-2000 and a 2 K warming, where already 70% of the simulations show significant changes (Fig. 5 d). As a result, the increase in low flows in the Alpine and Northern regions could, e.g., in combination with increased future annual precipitation in the GCMs (see Fig. 4), lead to a higher hydropower potential. On the contrary, a further decrease of available water (in low flows as well as annual precipitation) in the Mediterranean may pose additional water stress in that area. Although human influences such as reservoir management or human water demand are not considered in this study, different regional adaptation options should be considered depending on whether the world warms 2 K or 3 K. This holds also true for the more pronounced warming between 1.5 and 3 K, (Fig. 5, panels e and f) where the regional changes in low flows as well as the robustness amplify compared to 2 and 3 K warming. These results also highlight the non-linear sensitivity of changes in low flows to different levels of global warming. For example with long-lasting infrastructure or long planning horizons, adaptation strategies should be put in place now, without waiting for the 3 K level to be reached or not.

Overall, the robustness in the change signal rises with increased temperature differences between the warming levels. Based on the results of the multi-model assessment conducted here, significant differences in low flows between the policy relevant 1.5 K and 2 K warming could not be identified. Little differences between these two warming levels have been observed because of the high variability among the GCM/HM simulations. The multi-model variability is further analysed in detail in the following section.

## 3.3   Uncertainty contributions from GCMs and HMs

To provide a comprehensive picture over uncertainties, the signal to noise ratio (SNR) is investigated additionally to the robustness of the change signal based on the Wilcoxon rank sum test presented in sections 3.1 and 3.2. Furthermore, the uncertainty contribution of the GCMs and HMs for different levels of warming is also investigated.

**Table 4.** Dimensionless uncertainty contribution of GCMs and HMs averaged over the stratified European regions described in section 2.3.

| Warming level | European regions | | | | |
|---|---|---|---|---|---|
| | Alpine | Atlantic | Continental | Northern | Mediterranean |
| GCM uncertainty | | | | | |
| 1.5 K | 27.3 | 25.8 | 35.2 | 31.2 | 31.4 |
| 2 K | 32.1 | 31.6 | 44.7 | 43.7 | 38.5 |
| 3 K | 52.1 | 32.9 | 48.3 | 63.4 | 31.3 |
| HM uncertainty | | | | | |
| 1.5 K | 26.7 | 19.8 | 21.1 | 31.9 | 25.0 |
| 2 K | 33.4 | 24.0 | 25.3 | 39.6 | 30.2 |
| 3 K | 55.6 | 31.4 | 31.1 | 55.1 | 34.7 |

Under a global warming level of 1.5 K and 2 K, large parts of Europe exhibit substantial uncertainty, expressed as the SNR (Fig. 6, panels a and b). It is estimated as the ensemble median divided by the ensemble inter-quartile range (Giuntoli et al., 2015). Using the inter-quartile range partly accounts for outliers in the ensemble simulations. The SNR is small for changes in low flows under a warming level 1.5 K and increases with further warming. These results are similar to the increasing changes
5    and robustness of the simulations with the increased warming level as also previously discussed in Figure 3. Under a global warming level of 1.5 K, the spatial patterns of SNR and robustness coincide between the different methods (Fig. 3 b compared to Fig. 6 a). Nevertheless, a direct comparison of the uncertainty patterns under higher levels of warming between SNR and robustness leads to different conclusions in some regions. As an example, large parts of Germany show a robust change under 2 K and 3 K warming (Fig. 3, panels d and f) whereas the SNR is smaller than 0.8 over the same regions indicating a high
10   uncertainty. This occurs because the Wilcoxon rank sum test is performed for each ensemble member separately, and the result is independent of the sign of change and absolute value. On the contrary, the SNR shows the uncertainty among the ensemble members and depends on the variability between those ensemble simulations. Additionally, thresholds selected for rejecting results or marking them as uncertain have greater influence on the presented results in both methods. This highlights that the uncertainty information conveyed strongly depends on the metrics selected to represent them. In other words, the robustness
15   indicates that most ensemble members project significant changes in Germany, but there is disagreement among them indicated by a low signal to noise ratio.

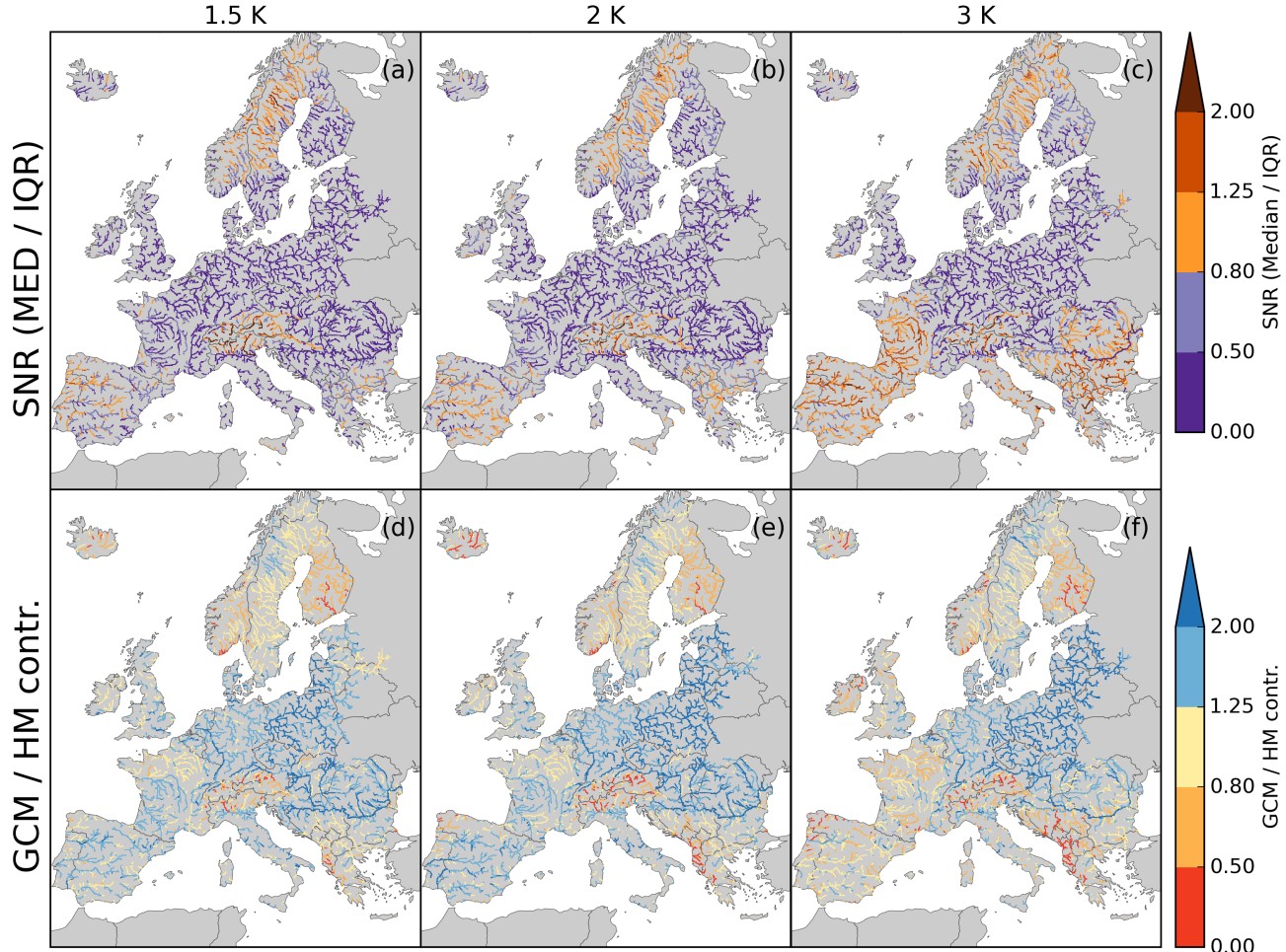

**Figure 6.** The upper row (a-c) shows the signal-to-noise ratio (ensemble median divided by the inter-quartile range) for the change in low flows (Q90) between the 1980s and 1.5 K (a), 2 K (b) and 3 K (c) warming. The relative uncertainty contribution of GCMs and HMs is shown in the lower row (d-f) for the three warming levels. Low values of GCM/HM indicate large HM uncertainty, values larger than one indicate a higher contribution of the GCMs to the total uncertainty.

The SNR results presented here are in line with the findings for the Rhine and Tagus River in Gosling et al. (2017). A comparison to other studies like Forzieri et al. (2014) or Roudier et al. (2016) is in this case difficult because those studies used different metrics to describe uncertainty and, consequently, the patterns in those studies vary significantly from the patterns shown here.

5   GCM and HM contributions to total uncertainty separated with the sequential sampling method (Samaniego et al., 2017a) are shown in Fig. 6 (d-f), and the spatially aggregated results over the IPCC Europe regions in Tab. 4. The uncertainty rises with higher levels of warming for both sources of uncertainty because of two reasons. The GCM uncertainty increases because a 30 year period reaching a 3 K warming often has a strong temperature period (with higher than average annual temperature)

within this 30 year period. On the contrary, GCM runs under the RCP2.6 often stabilise around a global warming level of 1.5 K. This pathway dependency of GCM runs influences the variability of the results with expectedly higher variability in the former case (James et al., 2017). The HM uncertainty increases with global warming because certain regions might cross thresholds. For example, parts of France might move from a energy-limited to a water-limited regime. The contribution of

the GCMs to the overall uncertainty across Europe is approximately 21% higher under a global warming level of 1.5 K, 25% higher under 2 K, and only 10% higher under a global warming of 3 K in comparison to the HM contribution. This decrease of GCM/HM contribution can be mostly attributed to the Mediterranean and Atlantic regions (in particular France). In these dry regions, the different representations of evaporation using temperature-based potential evapotranspiration used in mHM and PCR-GLOBWB will lead to a different evaporative response compared to explicitly solving the full energy-balance of

the land surface as in Noah-MP. Furthermore, HMs contribution to the total uncertainty is regionally higher than average in the Alpine and Northern regions, where snow accumulation and melt play an important role (Fig. 6 panels d-f). Snow processes are treated differently between the HMs, which explains the relatively high uncertainties in the Northern and Alpine area. Both mHM and PCR-GLOBWB use a temperature based conceptual degree-day method for snow processes, whereas the NOAH-MP model employs an energy balance scheme to resolve the snow accumulation and melt processes. In the Atlantic and

Continental regions, GCM uncertainty is higher under all levels of warming. One reason is that the lower quantiles of summer precipitation in CMIP5 simulations are generally underestimated and have a large spread in Central Europe (Liu et al., 2014). In the Rhine River basin, the spread in summer precipitation across the five GCMs used in this study is highest compared to other seasons (Krysanova and Hattermann, 2017). Remarkably, within the summer season the spread was higher under RCP8.5 compared to RCP2.6. Furthermore, HMs generally show nearly similar skill in humid areas where most of the models have

been developed and calibrated (Huang et al., 2017). The Northern area shows a nearly similar contribution in GCMs and HMs. In the Mediterranean, the uncertainty due to the HMs rises with increased warming. Reasons for such behavior could be the increased importance of the soil moisture and resulting actual evapotranspiration as well as infiltration treatment, which differs substantially between the HMs. For example, mHM uses separate storages for actual evapotranspiration and different runoff components (fast and slow interflow and baseflow components), whereas actual evapotranspiration and runoff depend on the

same storages in Noah-MP leading to a higher inter-variable dependency. This suggests that differences in soil and runoff representations within a model can have a significant effect on the simulation of future low flows, and can have a significant impact on the trend signal, as also had been previously noted by Wanders and Van Lanen (2015).

The procedure to differentiate between GCM and HM uncertainty has previously been presented in Samaniego et al. (2017a). They used six HMs forced with bias-corrected outputs from five GCMs under two RCPs set up in seven large river basins

worldwide for the period 1971-2099. Similar to the findings of this study, they also reported that uncertainty for a runoff index increases with time which corresponds to increased warming. Furthermore, the GCMs generally dominate the HMs uncertainty in low flows. Nevertheless, they also agree on the fact that the uncertainty contribution of the HMs depends on the hydro-climatic regime. Similarly, Vetter et al. (2015) used the ANOVA method to distinguish between different sources of uncertainty, including RCP uncertainty, which is not separately investigated here. For low flows, they came up with a 70%

contribution of RCPs on the drought impacts, with RCP uncertainty rising until the end of the 21st century. This may be ex-

plained due to the widening temperature range in the RCPs over time, which is not comparable to our approach of using a time sampling approach to identify different warming levels (Collins et al., 2013).

Overall, the regions showing higher uncertainty contribution from GCMs exhibited comparably lower SNR, indicating a significant variability in the GCM projections that are propagated through the HMs to the low flow signal. Furthermore, the contribution of the GCMs to the total uncertainty is higher than the contribution of HMs over Europe. Nevertheless, the influence of HMs cannot be neglected and outperforms the uncertainties in GCMs in some regions and depending on the warming level. Our results therefore strongly suggest the use of multiple hydrologic models for climate change impact assessment studies for future low flow projections, and that the use of single hydrologic models may provide misleading results.

## 4   Summary and Conclusions

Climate change is projected to alter low flows expressed as the Q90 indicator in Europe under a global warming of 1.5, 2 and 3 K. The magnitude of changes as well as the robustness in 45 member multi-model ensemble is amplified with increased levels of warming. Higher levels of warming therefore demand more distinctive adaptation actions. The mountainous regions in Europe show the strongest low flow increase from 22% under 1.5 K to 45% under a warming of 3 K. Continental Europe faces slight decreases in low flows. Higher decreases are expected in the Mediterranean (up to -35% under 3 K warming) and the Atlantic. We conclude that a warming level of 3 K will impose higher water stress over a large part of the Mediterranean, an area which already suffers from limited water resources that makes adaptation necessary. Further limitations in water availability may result in new managing challenges for water resource managers and policy makers, including the management of competition for water resources between sectors.

The projected changes in Q90 across Europe between the reference period (1971-2000) and a warming level of 1.5 K, as well as between a global warming of 1.5 K and 2 K are generally small with a low robustness and a small signal to noise ratio. It is not possible to distinguish climate impacts between a global warming of 1.5 K and 2 K. Nevertheless, some hotspot regions show changes greater than ± 10% between all warming levels investigated in this study. It would be misleading to conclude that mitigation of greenhouse gases is not needed. It is revealed here that large parts of the change in the climate induced low flow signal between the reference period and a global warming level of 2 K already happens before reaching the warming level of 1.5 K, specifically in the Alpine regions, Northern Europe and the Mediterranean. Therefore, mitigating climate change even below the 1.5 degree goal (UNFCCC, 2015) would be necessary to reduce negative drought impacts in hotspot regions like the Mediterranean.

The results shown here are independent of the uncertainty in emission scenarios. On the other hand, the uncertainty of the determination in the time periods for different warming levels is introduced. Generally, the robustness in the simulations and signal to noise ratio in the ensemble rise with increased warming and with the magnitude of change. As a result, regions with relatively large changes in low flows show a relatively low uncertainty in the results and have therefore the highest need to adapt to changing conditions. It is observed here that the selection of metrics to define uncertainty strongly influences the result. Here, we use the combination of robustness covering the significance in the change for every single ensemble member together

with SNR pointing at the variability and strength of the signal for the overall ensemble. Uncertainties should be considered in adaptation planning, e.g., in deciding to use climate impact simulations to determine regional vulnerability quantitatively or qualitatively. We conclude that the combination of different kinds of information, namely the change signal, the robustness and SNR, should be used in the adaptation process. These can be used to decide e.g., on the adaptation need or if a quantitative or

qualitative approach should be chosen for the estimation of regional vulnerability to climate change.

It is observed that the GCMs contribution to the overall uncertainty is higher than the HM contribution across Europe and that the HM contribution to total uncertainty rises with increased warming. This is related to the exhibited strong correspondence between the changes in mean annual total precipitation and streamflow Q90, which is strongest in lower warming levels and in the Atlantic and Continental Europe. Nevertheless, the HM contribution cannot be neglected and in some regions, it is

higher than the GCM contribution especially in the Alpine, Northern and Mediterranean, with rising global temperatures. The main reasons are the rising importance of hydrologic process description of snow, soil moisture and evapotranspitation, and infiltration. We conclude that climate change studies focusing on river low flows should employ large multi-model ensembles including multiple driving climate models as well as multiple impact models to provide a comprehensive analysis of model uncertainty.

*Competing interests.*   The authors declare that no competing interests are present.

*Acknowledgements.*   This study has been funded within the scope project HOKLIM (www.ufz.de/hoklim) by the German ministry for education and research (BMBF, grant number 01LS1611A). This study has been partially funded by the Copernicus Climate Change Service. The European Centre for Medium Range Weather Forecasts implements this service and the Copernicus Atmosphere Monitoring Service on behalf of the European Commission. We would like to thank all the colleagues contributing to the EDgE (http://edge.climate.copernicus.eu/)

project. Furthermore, we acknowledge the funding from NWO Rubicon 825.15.003. The ENSEMBLES data used in this work was funded by the EU FP6 integrated project ensembles (contract number 505539) whose support is gratefully acknowledged. We acknowledge the E-OBS dataset from the EU FP6 project ENSEMBLES (http://ensembles-eu.metoffice.com) and the data providers in the ECA&D project (http://www.ecad.eu). We would like to thank people from various organizations and projects for kindly providing us the data which are used in this study, which includes ISI-MIP, JRC, ESA, NASA, USGS, GRDC, BGR, UNESCO, ISRIC, EEA, EWA & CEDEX.

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
