# Peer review of "Climate change alters low flows in Europe under a global warming of 1.5, 2, and 3 degrees"

_Hydrology and Earth System Sciences, 2017_

## Referee Comment (RC1) · Anonymous Referee #1 · 3 Sep 2017

The authors present a comprehensive study of change in low flows for Europe using downscaled GCM output fed into three different hydrologic models. I am happy to recommend publishing of the manuscript subject to maybe some clarifications.

**This paper is looking at changes in the percentile (as the abstract says) – but the introduction is focuses on droughts. As it is currently phrased I am not sure I feel comfortable with research Question 1. I think this should be changed to say it is looking at changes in low flows. The introduction needs some text to relate drought to low flows. I understand that at the bottom of page 4 it is stated that Q90 is the drought metric but this comes too late in the piece.**

**I think there are a few papers that could be cited in the introduction, for example, Hall et al. (2014); 10.5194/hess-17-325-2013; and a recent article that looks at the**

[Figure]

sensitivity of flows to temperature 10.1038/s41598-017-08481-1.

**I did find it odd that a lot of material was introduced in the discussion on Page 10 and Page 17/18. Given it is relevant I think the introduction needs to (at least briefly) incorporate these references to put this works novelty in context.**

**Could the bias correction be elaborated in a sentence or two because the choice of bias correction can make a huge difference to the results? Especially if the focus is drought, authors need to correct for low-frequency variability biases - see 10.1016/j.jhydrol.2016.04.018.**

**Worth noting we are tracking for higher increases than 3 degrees probably: 10.1038/nclimate1783**

**Can the results in Table 1 be verbally contrasted with land predictions for Europe (i.e. will Europe heat up more or less than the global average). The IPCC reports will have this.**

**I am pretty sure that the low flow statistics in Table 2 are based on average of all the grid cells in a region but I am not sure. This could be mentioned in the text.**

**Figure 4 – not really clear to me what the blue dashed line indicates. I think the lines need to be described in the legend.**

**It is a bit hard to assess Table 3 because the step changes aren't linear. You could compare the following: Table1 Row 1 (0-1.5K) increase equivalent to 22, -7, -4, 8, -12 % changes and comparing to Row 3 in Table 2 (again a 1.5 K increase but now from 1.5 to 3K) of 24, -13, -12, 23, -23.**

**It was not clear to me how the GCM and HM signal-to-noise ratio was split.**

**Abstract Line 5: Unprecedented is a strong word and I would remove it.**

**Page 8 Line 4: Typo. ". . .by first fixing a HM and then calculating the range of Q90 (max-min) corresponding to give GCM outputs and repeating the previous step . . ."**

---

## Referee Comment (RC2) · Anonymous Referee #2 · 27 Sep 2017

General comments

This manuscript explores the impact of climate change on low river flows in Europe using a multi-model GCM and hydrological model (HM) ensemble under three global warming scenarios. The use of this ensemble allows the authors to assess the range of uncertainty in projections and the relative contributions of GCMs and HMs. Overall, it is an interesting and informative study, well-written and clear, supported by appropriate figures and references. There are some questions surrounding catchment selection for model validation and the general omission of smaller catchments, as well as the extent to which conclusions can be drawn on drought when analysing only flow percentiles. However, once these and some other interpretational aspects are addressed, I would recommend this study for publication.

[Figure]

Specific comments

Evaluating model performance (Page 4, line 30 to Page 5, line 4): I think more interpretation is required of Fig 1. There are only nine lines devoted to this, and I am not sure that I entirely agree with the assessment that "results show a good agreement" without some caveats. Low flows for PCR-GLOBWB and median flows for Noah-MP are systematically over-estimated across almost all catchment sizes, and there is a systematic under-estimation of low flows for Noah-MP. Whilst no-one is expecting perfect model results, there should be more attention given to the validation, as well as additional text in the discussion on the potential influence of model performance on the conclusions drawn.

Catchment selection for validation (Figure 2): There is no information on how or why these catchments were selected for validation. It would appear that a number of nested sub-catchments of relatively few large rivers have been selected (i.e. multiple downstream stations on the Rhone, Loire, Ebro, etc.) There is also no information on from where the river flow data were sourced. Data are freely available for some regions where the models are not evaluated but for which results are presented.

Omission of catchments <10,000km2 (Page 8, lines 13-15): Perhaps this argument explains the selection of catchments in Fig 2? I am not convinced that modelled data at 5km spatial resolution cannot resolve the river flow network of catchments <10,000km2. The authors highlight the "unprecedented" (Abstract) 5km spatial resolution and on a number of occasions highlight the "spatially explicit information" in this study, but removing smaller catchments seems not to capitalise on this. This section also says that such catchments will be removed, but the maps displayed in Fig 3 onwards all feature a river flow network which contains routed flows for catchments less than 10,000km2, in which the network appears to be relatively well defined. All of this is relevant also in relation to the comment above on model performance at the lower end of the flow regime across all HMs (Fig 1). Catchments <10,000km2 also omitted from Fig 4; are the results similar?

Drought or low flows (throughout manuscript): There is some inconsistency between the use of 'drought' and 'low flows'. This paper analyses changes in median annual Q90 flows, which allows conclusions to be drawn on climate change impacts on low flows but not necessarily drought. The authors use low flows and drought at times interchangeably, including in the research questions and conclusions.

Robustness (Page 10, line 6): There is detail on the hotspots of changes in low flows, but in the end the low robustness means that for the Mediterranean / Atlantic, changes are not 'likely' (as defined by the authors) for most of these areas for either 1.5K or 2K. In fact, the signal for the Mediterranean might be stronger than that for the Atlantic, but it is less robust than the Atlantic. Statements like "Nevertheless, these results are not robust" (Page 13, lines 17-18) could be useful here.

Uncertainty from GCMs or HMs: There are a number of statements on Page 18 that need to be clarified in relation to Table 4. "HMs are the major source of uncertainty in the Alpine region" – GCMs and HMs are closer together in Alpine compared with other regions, but the numbers in Table 4 are similar for GCMs and HMs across all warming levels, and GCMs are higher for 1.5K. "The Northern area shows a nearly similar contribution in GCMs and HMs" – so does Alpine (see above), and GCMs and HMs are even more comparable for 2K and 3K in Alpine than in Northern. "In the Mediterranean, the uncertainty due to the HMs rises with increased warming" – this is true for all regions. It is also strong to say that GCMs "dominates" total uncertainty for Europe (Page 18, line 33), especially given the negligible differences between GCMs and HMs for two of the five regions.

Technical corrections

Page 2, line 32: "differ- ent" to "different"

Page 4, line 11 - Page 5, line 4: Very lengthy paragraph could be better structured and split into multiple shorter paragraphs.
Figure 1: Useful to have a legend for colour based on GCM as there are some systematic patterns.

Page 11, line 9: "Q10" should be "Q90"?

Page 11, line 11: "to a large extent"

Page 15, line 11: Mediterranean should be "(-16%)" not "(-24%)", reading off Table 3 for 2K to 3K?

Page 17, line 1 (and throughout): "Targus" should be "Tagus"?

Table 4: It's more editorial, but Fig 6 discussed before Table 4 despite being featured afterwards.

---

## Referee Comment (RC3) · Anonymous Referee #3 · 29 Sep 2017

This manuscript deals with a multi-GCM and multi-hydrological models assessment of changes in low flows across Europe between a present-day period (1971-200) and 3 different global warming levels: 1.5K, 2K, and 3K (and between them as well). It therefore contributes to document the effects of climate change on low-flow hydrology in Europe in the context of the Paris Agreement. This manuscript thus deals with a topical and important topic, and fits well into the scope of HESS. It is generally well structured and written, and conclusions are generally well supported by results shown. I have however two main comments (as well as specific comments) detailed below that should be addressed before the manuscript is published in HESS.

[Figure]

**1 Main comments**

**1.1 Hydrological calibration and simulation over influenced catchments**

The calibration details (specific comment #9 and #10) as well as the validation results (specific comments #12, #13, #14) do not give enough confidence on the quality of hydrological modeling, and highlights the issue of calibrating and/or validating – seemingly natural-catchment-only – models against highly influenced catchments like the Ebro or the Rhône, especially for low flows. First there is not enough information on the calibration process, and even the catchments used for that are not identified. Second, validation is done for a large part over influenced catchments, and also over ensembles of highly nested catchments. Both points should be reconsidered in a future revision of the manuscript.

**1.2 Scale of catchments selected for presenting and averaging results**

There are numerous inconsistencies throughout the manuscript in terms of the minimum catchment size used for presenting results (and giving averaged figures), see specific comments #20, #21, #22, #24, #27. Addressing this comment may imply reformatting all results, but this is also intrinsically linked to main comment #1. Indeed, the manuscript state that the runoff routing scheme prevent using results for catchments smaller than 10000 km2, and near-natural catchments are usually only smaller than that in Europe. In parallel, maps of results are given over a river network encompassing drained areas much small than the indicated threshold. This thus shades doubts (maybe unjustified, but is has to be demonstrated) on the validity of models and results, together with issues highlighted in main comment #1.

**2 Specific comments**

1. P1L2, "1.5, 2 and 3 K": please specify that this is with respect to the preindustrial period

2. P1L10, "-12%": What is the baseline period here? This is all the more important that there could easily be confusion with the baseline used for the global warming level (see above).

3. L11-12: this sentence is ambiguous. Less snowmelt may imply less streamflow in some conditions (e.g. constant liquid precipitation or declining total precipitation). Please rephrase and make it clearer.

4. P1L13-14: This sentence is also quite ambiguous. What is exactly preventing distinguishing between 1.5 and 2K warming effects? Is it the interannual variability which prevents distinguishing statistically estimates of period-averaged changes-for a given GCM-HM combination? Or is it the uncertainty due to the multimodel ensemble that prevents distinguishing ensembles of multimodel period-average estimates between present and future? Or both? Please make it clear here.

5. P2L11: The low-flow component of the 2015 drought event has been specifically studied by Laaha et al. (2017). I believe this reference is worth adding to the manuscript.

6. P2L21-27: What is the time slice that corresponds to the quantitative and qualitative results recalled here? Please make it clearer.

7. P4L7-10: First, this interpolation step should not be called downscaling as the latter refers to methods that actually add information for each day (either through regional climate models or empirical-statistical downscaling models) to the larger scale GCM fields. I would therefore strongly recommend using "disaggregating" or "disaggregated" instead of "downscaling" or "downscaled" in the manuscript.

8. P4L7-10: Second, this interpolation step should be better documented here, in order for the reader to understand the advantages and shortcomings of this approach, which are essential for assessing the quality of subsequent hydrological simulations. This interpolation step should ideally be assessed using a global reanalysis against high-resolution gridded datasets, like RCMs are actually assessed (see e.g. Kotlarski et al., 2017, among many others, for a recent example). This would critically allow distinguishing errors coming from (1) the spatial interpolation technique (and their large-scale forcings), and (2) the hydrological models. Please at least add some comments on that in the manuscript. Plus, the reference used for this interpolation technique is incomplete in the list of references.

9. P4L24: What are these 9 catchments? Please provide some more information (location, surface, etc.). Are they near-natural or influenced catchments?

10. P4L24-26: What is the period used for calibration? And what are the calibration criteria (for both automatic and manual calibration)? Are they specific for low flows? Please carefully specify all this in the manuscript.

11. P4L27-29, "The assessment. . . (Gosling et al., 2017)": This is a very strong statement, which I tend to disagree with at least as a general conclusion. This is moreover hardly supported by the reference given in the manuscript, which compares global hydrological models and catchment hydrological models for the Rhine and Tagus (and other catchments, but not located in Europe). Results for a low flow indicator (Q95) show a large divergence of the two types of models with increasing global warming level (Gosling et al., 2017, their Fig. 2). As a conclusion, I would therefore strongly recommend removing this statement from the manuscript.

12. P4L33-P5L4 and Figure 1: The assessment of HMs is very light and not strongly supported by Fig. 1. Indeed, this figure is potentially misleading, as it basically only checks that catchments have equally small/large indicators (Q90 or Q50)

for both observations and simulations, which is mainly driven by the size of the catchment. I would therefore recommend using a different and more informative representation of differences, preferably in terms of relative errors (in percents), and also preferably as maps in order to show the potential spatial pattern in errors. This representation would also greatly help in comparing present-day errors to relative changes presented later in the manuscript. I personally would not give too much credit for a model showing for a given location present-day errors as large as 3K future changes. . .

13. Figure2, right: This figure shows the location of validation gauges used in Fig. 1. First, it shows that many points in Fig. 1 comes from the same rivers and are necessarily highly correlated, which inherently bring some bias to the results that should be representative of the whole Europe. I would strongly recommend removing redundant points scatter plots like presented in Fig. 1. This would not be a problem however with suggested spatial representations (cf. above).

14. Figure 2, right: The second point is that several validation gauges are located on highly influenced rivers. For example, the Ebro river (Spain) is heavily influenced by water abstractions for irrigation, and the seasonal regime of the Rhône river (France) is heavily influenced by all the hydropower reservoirs located in the Alps (and other surrounding mountain ranges). There are many other cases that can be spotted on the map. As a consequence, observed streamflow indicators for low flows simply cannot be compared to natural (i.e. without human influence) hydrological simulations for these catchments. A good fit to observations may indeed reveal that physical parameters in HMs are tweaked to compensate for no representation of human influence. This may not be a problem in itself (at least for practical modeling purposes if not scientifically satisfactory) if human influences would not have changed and would not change in the future. Which has happened and definitely will. As a conclusion, I would strongly recommend using only near-natural catchments as validation (and also calibration) gauges for

natural hydrological modeling (as I suppose it is the case in the manuscript, even if some HMs considered may represent human influences). A number of reference hydrometric networks have recently been developed at the country scale (Hannaford and Marsh, 2008; Giuntoli et al., 2013; Murphy et al., 2013), and one should take advantage of these. Note that these networks overlap for some countries (but not for some other) with stations tagged "climate sensitive" in the Global Runoff data Centre.

15. P6L3: The 0.46K figure has uncertainties attached to it, according to the reference cited (Vautard et al., 2014). Please do mention these uncertainties in the manuscript, with possibly additional references that provides 1971-2000 estimates of global warming level.

16. P6L20: The use of calendar year is not entirely satisfactory for computing Q90 in snow-influenced catchments where the low-flow period (or one of the low-flow periods, which is a more difficult situation) may span two calendar years. Please consider changing the calculation procedure or at least justify this approximation.

17. P7L8-9: Please mention here (rather than in the results section) that the robustness is compute as the percentage of projections showing a significant change.

18. Table 2: Please make clear that "1980s" refers to the 1971-2000 period.

19. P8L3-9: I don't really understand this peculiar choice of method for computing the relative contributions of uncertainty from GCMs and HMs. Many studies demonstrated that simple Analysis of Variance (ANOVA) approaches are perfectly suited to this case, and it has been recently widely applied to compute contribution from GCMs and HMs (see e.g. Giuntoli et al., 2015; Vetter et al., 2017, among many others), even by some of the authors of the present manuscript (Mishra et al., 2015). ANOVA approaches can critically take account of GCM/HMs interactions, which is presumably not the case of the method used, and of the different sizes

of fixed effects. The set-up is here rather simple compared to more complex ones that consider unbalanced number of runs from each GCMs and/or multiple sources of uncertainty (see e.g. Addor et al., 2014; Vidal et al., 2016). I therefore strongly recommend using a simple two-way ANOVA approaches for the present study, or at least check current results against a simple two-way ANOVA approach. Indeed, I am unsure of how this sequential sampling approach relates to the more traditional ANOVA approach, and what their respective underlying hypotheses are. I would welcome some online discussion on this.

20. P8L13-15: First, this should come much earlier in the manuscript. Second, this is not consistent with maps of streamflow changes that seemingly include results for catchments with a surface lower than 10000 km2. This should be clarified. This is closely linked to specific comment #14.

21. Figure 3 (and Fig. 5 and Fig. 6). See comment above. Plus, the figure indicated above each map is seemingly a continental average of the plotted value along the river network. First, this should be clarified. Second, this value is closely related to the choice of the minimal catchment surface area considered. Values would be very different if, as stated P8L13-15, only catchments with an area larger than 10000 km2 would be considered. Please make all these statement and results consistent across the manuscript.

22. P10L3-4: This statement is somewhat inconsistent with the choice of the calendar year use for the calculation of Q90. Please clarify this in the manuscript.

23. P10L7, "models": I presume this should be "simulations".

24. P11L1, "new spatially explicit information". This is again contradictory with the 10000 km2 statement. Cf. comments above.

25. P11L16-17.This sentence is ambiguous. The increased spread along the 1:1 line (i.e. when smaller and larger values are considered) does indeed contribute to
a higher coefficient of determination, which is not the case for the spread across (i.e. with higher residuals from) the 1:1 line. Please rephrase.

26. Figure 4: Several presumably regression lines are given on the graph. Please either define and comment them, or remove them. Also, please add lines delimiting the quadrants.

27. Figure 4: The legend states that only catchments with a surface area higher than 10000 km2 are considered. This is again not consistent with values provided by other figures.

28. P13L3-5: This is already written P10L35-P11L2. And this is commented in specific comment #24.

29. Title of Section 3.2: The difference between section 3.1 and section 3.2 are not understandable based on this title, and the reader may be unsettled at this point as I was. There should be something of a "between the levels of warming" somewhere. Please rephrase.

30. Figure 5. Cf. comment #21.

31. P15L15-16: The increase in winter low flows would not necessarily lead to a higher hydropower potential. It actually depends on the evolution of total precipitation. And the possible evolution of hydropower production would depend on the type of reservoir management, as well as management rules constrained by possible other water usages (sustaining summer low flows downstream, irrigation, recreation, etc.). Moreover, a decrease in low flows does not necessarily imply a decrease in overall water availability average over the year, and the water stress is conditional on the respective weight of water availability and water demand for a given time. So I would recommend adapting the statements according to the above comments.

32. P15L17-18: I however completely agree with the need of regional adaptation options. Except that adaptation strategies should be put in place now, without waiting for the 3K level to be reached or not.

33. P16L6, "the result is independent of the sign of change": Well, this is a potentially serious issue. Indeed, how to interpret a situation where e.g. out of15 projections, 5 give a significant upward change, 5 other no significant change, and the last 5 a significant downward change? I would recommend interpreting this situation with particularly no robust signal! So please make clearer in the manuscript all the different possible cases and the way to interpret them. An alternative for presenting robustness would be the one used in the IPCC AR5 WGI report, i.e. the percentage of projections agreeing on the sign of the change.

34. P16L11-13: I totally agree with this sentence, but it comes here out of the blue. Please consider moving it to the introduction, discussion, or conclusion.

35. Figure 6. The choice of colour breaks is here particularly unfortunate here. For the SNR, I would appreciate having a break in value 1, in order to see where the median change is higher than the uncertainty in projections. For the ratio of GCM to HM uncertainty contribution, this is all the more important to see where this crosses the 1 value. An alternative would be to use bivariate colour scales (Teuling et al., 2011) to jointly plot the evolution of both sources of uncertainty.

36. P17L4-5: This exact sentence has already been written P8L3-4, and commented above (comment #19)

37. P17L8-P18L3: I am more or less OK with what is written here, but I do not understand why this would imply that the ratio of HM contribution to GCM contribution is higher at the 3K level. Please provide some explanations in the manuscript. Couldn't this be related to timing of threshold crossing in HM behavior that would

differ from one HM to another, e.g. going from energy-limited to water-limited evaporation process?

38. P18L4-20: This whole paragraph tends to support the above hypothesis. This should be related in the manuscript to recent uncertainty decompositionresults obtained for a catchment located in the Southern Alps. It showed that the increasing spread of changes in future low flows by different HMs is linked to increasing spread in simulated evaporation and snow water equivalent (Vidal et al., 2016).

39. P19L28-30, "We conclude... support the adaptation process." Well, this is actually only a wish. Nothing in the paper allows asserting that, even I personally hope this is the case. So please rephrase.

**3 Technical corrections**

1. P1L5, "unprecedented": it is a bit far-fetched, given that (1) GCM forcings are only disaggregated to this resolution without adding any downscaling information, and (2) results are seemingly partly given only for catchments >10000 km2 (P8L13-15).

2. P1L6: "combination"

3. P2L2: "independently"?

4. P2L22-24: I believe that the sentence is not grammatically correct.

5. P2L30: "2"? in reference (UNFCC, 2015)

6. P2L34: "because of"

7. P3L3: please check missing or incorrect "the"

8. P3L8: "southern Europe"

9. P11L11: "extent"

10. P13L5: "political" -> "policy". Also P15L23.

11. P15L22, "distinguished": please rephrase.

12. P15L24, "ensemble members": Please clarify what they are.

13. P20L1, "pronounced": What is? Please rephrase.

14. P21L5-8: Wrong formatting, cf. IPCC report citation rules.

15. P23L34: line feed

16. P24L25-26: extra information to be removed

**4  References**

Addor, N., Rössler, O., Köplin, N., Huss, M., Weingartner, R.  Seibert, J. (2014) Robust changes and sources of uncertainty in the projected hydrological regimes of Swiss catchments.  Water Resources Research, 50(10), 7541-7562.  doi: 10.1002/2014WR015549

Giuntoli, I., Renard, B., Vidal, J.-P.  Bard, A. (2013) Low flows in France and their relationship to large-scale climate indices. Journal of Hydrology, 482, 105-118.  doi: 10.1016/j.jhydrol.2012.12.038

Giuntoli, I., Vidal, J.-P., Prudhomme, C.  Hannah, D. M. (2015) Future hydrological extremes: the uncertainty from multiple global climate and global hydrological models. Earth System Dynamics, 6(1), 267-285. doi: 10.5194/esd-6-267-2015

Gosling, S. N., Zaherpour, J., Mount, N. J., Hattermann, F. F., Dankers, R., Arheimer, B., Breuer, L., Ding, J., Haddeland, I., Kumar, R., Kundu, D., Liu, J., van Griensven, A., Veldkamp, T. I. E., Vetter, T., Wang, X. Zhang, X. A comparison of changes in river runoff from multiple global and catchment-scale hydrological models under global warming scenarios of 1.5°C, 2°C and 3°C (2017) Climatic Change, 141(3), 577-595. doi: 10.1007/s10584-016-1773-3

Hannaford, J. Marsh, T. J. (2008) High-flow and flood trends in a network of undisturbed catchments in the UK. International Journal of Climatology, 28(10), 1325-1338. doi: 10.1002/joc.1643

Kotlarski, S., Szabó, P., Herrera, S., Räty, O., Keuler, K., Soares, P. M., Cardoso, R. M., Bosshard, T., Pagé, C., Boberg, F., Gutiérrez, J. M., Isotta, F. A., Jaczewski, A., Kreienkamp, F., Liniger, M. A., Lussana, C. and Pianko-Kluczyńska, K. (2017) Observational uncertainty and regional climate model evaluation: a pan-European perspective. International Journal of Climatology, in press, doi:10.1002/joc.5249

Laaha, G., Gauster, T., Tallaksen, L. M., Vidal, J.-P., Stahl, K., Prudhomme, C., Heudorfer, B., Vlnas, R., Ionita, M., Van Lanen, H. A. J., Adler, M.-J., Caillouet, L., Delus, C., Fendekova, M., Gailliez, S., Hannaford, J., Kingston, D., Van Loon, A. F., Mediero, L., Osuch, M., Romanowicz, R., Sauquet, E., Stagge, J. H. Wong, W. K. (2017) The European 2015 drought from a hydrological perspective. Hydrology and Earth System Sciences, 21(6), 3001-3024. doi: 10.5194/hess-21-3001-2017

Mishra, V., Kumar, R., Shah, H. L., Samaniego, L., Eisner, S. Yang, T. (2017) Multimodel assessment of sensitivity and uncertainty of evapotranspiration and a proxy for available water resources under climate change. Climatic Change, 141(3), 451-465. doi: 10.1007/s10584-016-1886-8

Murphy, C., Harrigan, S., Hall, J. Wilby, R. L. (2013) Climate-driven trends in mean and high flows from a network of reference stations in Ireland. Hydrological Sciences Journal, 58(4), 755-772. doi: 10.1080/02626667.2013.782407

Teuling, A. J., Stöckli, R. Seneviratne, S. I. (2011) Bivariate colour maps for visualizing climate data International Journal of Climatology, 31(9), 1408-1412. doi: 10.1002/joc.2153 Vautard, R., Gobiet, A., Sobolowski, S., Kjellstrom, E., Stegehuis, A., Watkiss, P., Mendlik, T., Landgren, O., Nikulin, G., Teichmann, C. Jacob, D. (2014) The European climate under a 2 degrees C global warming. Environmental Research Letters, 9(3). doi: 10.1088/1748-9326/9/3/034006

Vetter, T., Reinhardt, J., Flörke, M., van Griensven, A., Hattermann, F., Huang, S., Koch, H., Pechlivanidis, I. G., Plötner, S., Seidou, O., Su, B., Vervoort, R. W. Krysanova, V. (2017) Evaluation of sources of uncertainty in projected hydrological changes under climate change in 12 large-scale river basins. Climatic Change, 141(3), 419-433. doi: 10.1007/s10584-016-1794-y

Vidal, J. P., Hingray, B., Magand, C., Sauquet, E. Ducharne, A. (2016) Hierarchy of climate and hydrological uncertainties in transient low flow projections. Hydrology and Earth System Sciences, 20(9), 3651-3672. doi: 10.5194/hess-20-3651-2016

---

## Author Comment (AC1) · 1 Nov 2017

author comments RC1 11

November 1, 2017

We thank the reviewer for the time and effort in commenting on our manuscript. We provide responses to each individual point below. For clarity, comments are given in normal font, and our responses are given as blue text.

The authors present a comprehensive study of change in low flows for Europe using downscaled GCM output fed into three different hydrologic models. I am happy to recommend publishing of the manuscript subject to maybe some clarifications.

**This paper is looking at changes in the percentile (as the abstract says) ? but the introduction is focuses on droughts. As it is currently phrased I am not sure I feel comfortable with research Question 1. I think this should be changed to say it is looking at changes in low flows. The introduction needs some text to relate drought to low flows. I understand that at the bottom of page 4 it is stated that Q90 is the drought metric but this comes too late in the piece.**

Thank you for this helpful comment. We agree to include the clear differentiation between the terms "hydrological drought" and "low flow" and we will adapt research question 1, accordingly. We suggest to include the paragraph:

"This study investigates low streamflow, defined as Q90, representing daily streamflow exceeding 90% of the time, which has the potential to impact hydrological drought.

Hydrological drought is associated with shortfalls on surface or subsurface water availability which can occur in low streamflow, groundwater, or reservoir levels. Changes in low flows shown in this study can, but will not in every case, result in drought. Exceptions are e.g. riverine based transportation, where streamflow values below a threshold level are defined as hydrological drought."

**I think there are a few papers that could be cited in the introduction, for example, Hall et al. (2014); 10.5194/hess-17-325-2013; and a recent article that looks at the sensitivity of flows to temperature 10.1038/s41598-017-81-1084.**
We agree to include 10.5194/hess-17-325-2013 in the introduction. Two other suggested paper are deemed beyond the scope of this manuscript.

**I did find it odd that a lot of material was introduced in the discussion on Page 10 and Page 17/18. Given it is relevant I think the introduction needs to (at least briefly) incorporate these references to put this works novelty in context.**
Thank you for your suggestion. We will extend introduction with the studies mentioned in the discussion.

**Could the bias correction be elaborated in a sentence or two because the choice of bias correction can make a huge difference to the results? Especially if the focus is drought, authors need to correct for low-frequency variability biases - see 10.1016/j.jhydrol.2016.04.018.**
There is a huge number of bias correction methods available, all facing advantages and disadvantages
(e.g. https://doi.org/10.1016/j.jhydrol.2012.05.052) for hydrological impact studies. The advantage of the method applied in our manuscript (Hempel et al.) is that it is trend preserving, which is of major importance for climate impact studies. We would have a different opinion, if this is meant by the comment that "authors need to correct

for low-frequency variability biases" in daily precipitation and temperature, because they are hardly directly linkable to low flow events.
We refer to the statement in in Donnelly et al (2017): "Cannon et al. (2015) and Maurer and Pierce (2014) showed that approaches like the quantile mapping used here can change the climate signal in the raw CM output significantly. Nevertheless, it is still unclear which methods give the most realistic climate change projections.".

**Worth noting we are tracking for higher increases than 3 degrees probably: 10.1038/nclimate1783**
Agreed.

**Can the results in Table 1 be verbally contrasted with land predictions for Europe (i.e. will Europe heat up more or less than the global average). The IPCC reports will have this.**
This is a good suggestion, but out of scope of this manuscript. Europe warms faster than the global mean, which has been visualised (for the underlying 5 GCM simulations) in http://edge.climate.copernicus.eu/Apps/#climate-change

**I am pretty sure that the low flow statistics in Table 2 are based on average of all the grid cells in a region but I am not sure. This could be mentioned in the text.**
We we will reformulate accordingly.

**Figure 4 ? not really clear to me what the blue dashed line indicates. I think the lines need to be described in the legend.**
Agreed. The dashed lines show two regressions (for positive and for negative deviations).

\# It is a bit hard to assess Table 3 because the step changes aren't linear. You could compare the following: Table1 Row 1 (0-1.5K) increase equivalent to 22, -7, -4, 8, -12 changes and comparing to Row 3 in Table 2 (again a 1.5 K increase but now from 1.5 to 3K) of 24, -13, -12, 23, -23.

For clarity, we suggest to include a row "absolute warming" (tab. 2: 1.04 K, 1.54 K, 2.54 K; tab. 3: 0.5 K, 1 K, 1.5 K). Comparison of results in tab. 2 and 3 rarely gives added value to the manuscript as changes are nonlinear with warming and regionally different - this is already reflected.

\# It was not clear to me how the GCM and HM signal-to-noise ratio was split.

The SNR was calculated for the combined GCM/HM runs (no splitting). If you refer to the GCM/HM uncertainty, the approach is described on page 8, line 2-8 in detail.

\# Abstract Line 5: Unprecedented is a strong word and I would remove it.

For Europe, there is no study available using a multi-model ensemble with 45 members including three impact models for low flows and at a high spatial resolution of $5\times 5$ $km^2$. We think this justifies the usage of the term "unprecedented".

\# Page 8 Line 4: Typo. "...by first fixing a HM and then calculating the range of Q90 (max-min) corresponding to give GCM outputs and repeating the previous step ..."

Agreed.

---

## Author Comment (AC2) · 1 Nov 2017

author comments RC2 11

November 1, 2017

We thank the reviewer for the time and effort in commenting on our manuscript. We provide responses to each individual point below. For clarity, comments are given in normal font, and our responses are given as blue text.

General comments

This manuscript explores the impact of climate change on low river flows in Europe using a multi-model GCM and hydrological model (HM) ensemble under three global warming scenarios. The use of this ensemble allows the authors to assess the range of uncertainty in projections and the relative contributions of GCMs and HMs. Overall, it is an interesting and informative study, well-written and clear, supported by appropriate figures and references. There are some questions surrounding catchment selection for model validation and the general omission of smaller catchments, as well as the extent to which conclusions can be drawn on drought when analysing only flow percentiles. However, once these and some other interpretational aspects are addressed, I would recommend this study for publication.

Thank you for the overall positive feedback. From the altogether three reviews, we realised that the information given on calibration and validation needs to be extended in the manuscript. Smaller catchments have not generally been omitted. The catchment size at a horizontal resolution of $5 \times 5$ km$^2$ is limited by the DEM in determining the

catchment boundaries. Therefore, the results and conclusions in this study are based on catchments (or better river grid cells) with a contributing area >1000 km$^2$ have been used for the study, and these are shown in figure 3 and figure 5.

The selection of catchments >10000 km$^2$ in figure 1 and 4 in the first version of the manuscript has been done for clarity reasons. This will be changed for validation figure 1. Notably, for the validation (see attached figure) we selected 357 basins based on daily streamflow data availability (selection criteria; complete dataset of 30 yrs, 1966-1995, this time period would change (old: 1971-2000) because it resulted into largest sample size). Their median basin area is 1680 km$^2$.

Furthermore, the terms "hydrological drought" and "low flow" will be better specified and used in a coherent way.

Specific comments

Evaluating model performance (Page 4, line 30 to Page 5, line 4): I think more interpretation is required of Fig 1. There are only nine lines devoted to this, and I am not sure that I entirely agree with the assessment that "results show a good agreement" without some caveats. Low flows for PCR-GLOBWB and median flows for Noah-MP are systematically over-estimated across almost all catchment sizes, and there is a systematic under-estimation of low flows for Noah-MP. Whilst no-one is expecting perfect model results, there should be more attention given to the validation, as well as additional text in the discussion on the potential influence of model performance on the conclusions drawn.

Thank you for pointing this out. We suggest to include the following paragraph on the calibration of the hydrological models using observed meteorological forcing data (which focussed on headwater catchments). We suggest to include the paragraph:

"The three HMs used in this study were calibrated in nine European focus basins located in Spain, Norway and UK, which were selected based on the consultation with the user groups within the EDgE project. Besides these, we also included three more central EU catchments (located in France and Germany) to represent diversity

in hydro-climatic regimes. All HMs parameters were calibrated such that the model simulations represent a range of hydrologic regimes, rather than tailored to any specific characteristics. This was done in a consistent manner so that the model simulations can be used for a range of indicators (including high, low, and average flows) within the EDgE project. We recognize that HMs could be calibrated to a specific streamflow characteristic (in this case to low flows), but this was not considered within this study. We also note that the HMs do not consider human management effects in this study which could have substantial effect during the low flow times - as a result constraining the model to any specific low flow characteristic can result in a biased simulations. Also due to the similar reason we may expect a relatively lower model skill in matching the observed low flow characteristic."

The text "results show a good agreement" was written behind the background that GCM data for the time period 1971-2000, which differs from the observed weather in that period, was used to drive the HMs for the validation against simulated Q90 values. We agree that the discussion should be extended. Furthermore, based on Reviewer 3, Figure 1 will be re-drawn using specific discharge to remove the basin-scale dependency of the data (see above and attached figure). We acknowledge the adressed systematic biases and see only limited influence on the future results and conclusions drawn. It is not possible to determine if a model that fits perfectly in the past is also able to produce perfect results under changed climate conditions. Consequently, it cannot be concluded that imperfect models are not useable for estimating future (relative) changes.

Catchment selection for validation (Figure 2): There is no information on how or why these catchments were selected for validation. It would appear that a number of nested sub-catchments of relatively few large rivers have been selected (i.e. multiple downstream stations on the Rhone, Loire, Ebro, etc.) There is also no information on from where the river flow data were sourced. Data are freely available for some regions where the models are not evaluated but for which results are presented.

Agreed. The selection is partly explained in the general comments answer. We selected 357 basins based on data availability (selection criteria; complete dataset of 30 yrs, 1966-1995). Their median area is 1680 km$^2$. Fig. 2 will be adapted (see attached figure).

Omission of catchments <10,000km2 (Page 8, lines 13-15): Perhaps this argument explains the selection of catchments in Fig 2? I am not convinced that modelled data at 5km spatial resolution cannot resolve the river flow network of catchments <10,000km2. The authors highlight the "unprecedented" (Abstract) 5km spatial resolution and on a number of occasions highlight the "spatially explicit information" in this study, but removing smaller catchments seems not to capitalise on this. This section also says that such catchments will be removed, but the maps displayed in Fig 3 onwards all feature a river flow network which contains routed flows for catchments less than 10,000km2, in which the network appears to be relatively well defined. All of this is relevant also in relation to the comment above on model performance at the lower end of the flow regime across all HMs (Fig 1). Catchments <10,000km2 also omitted from Fig 4; are the results similar?
This is not the case, explanations have been given in the author replies above.

Drought or low flows (throughout manuscript): There is some inconsistency between the use of 'drought' and 'low flows'. This paper analyses changes in median annual Q90 flows, which allows conclusions to be drawn on climate change impacts on low flows but not necessarily drought. The authors use low flows and drought at times interchangeably, including in the research questions and conclusions.
Thank you for this helpful comment. We agree do include the clear differentiation between the terms "hydrological drought" and "low flow" and we will adapt research question 1, respectively. We suggest to include the paragraph:
"This study investigates low streamflow, defined as Q90, representing daily streamflow

exceeding 90% of the time, which has the potential to impact hydrological drought. Hydrological drought is associated with shortfalls on surface or subsurface water availability. These can occur i.e. in low streamflow, groundwater, or reservoir levels. Changes in low flow shown in this study can, but will not in any case, result in drought. Exceptions are e.g. riverine based transportation, where streamflows below a threshold level are defined as hydrological drought."

Robustness (Page 10, line 6): There is detail on the hotspots of changes in low flows, but in the end the low robustness means that for the Mediterranean / Atlantic, changes are not 'likely' (as defined by the authors) for most of these areas for either 1.5K or 2K. In fact, the signal for the Mediterranean might be stronger than that for the Atlantic, but it is less robust than the Atlantic. Statements like "Nevertheless, these results are not robust" (Page 13, lines 17-18) could be useful here.
We will modify the text as suggested.

Uncertainty from GCMs or HMs: There are a number of statements on Page 18 that need to be clarified in relation to Table 4. "HMs are the major source of uncertainty in the Alpine region" – GCMs and HMs are closer together in Alpine compared with other regions, but the numbers in Table 4 are similar for GCMs and HMs across all warming levels, and GCMs are higher for 1.5K. "The Northern area shows a nearly similar contribution in GCMs and HMs" – so does Alpine (see above), and GCMs and HMs are even more comparable for 2K and 3K in Alpine than in Northern. "In the Mediterranean, the uncertainty due to the HMs rises with increased warming" – this is true for all regions. It is also strong to say that GCMs "dominates" total uncertainty for Europe (Page 18, line 33), especially given the negligible differences between GCMs and HMs for two of the five regions.
Agreed. We will modify the text as suggested.

Technical corrections

Agreed. We will implement them in the revised manuscript.

Page 2, line 32: "differ- ent" to "different"

Page 4, line 11 - Page 5, line 4: Very lengthy paragraph could be better structured and split into multiple shorter paragraphs.

Figure 1: Useful to have a legend for colour based on GCM as there are some systematic patterns.

Figure 1 will be re-drawn (attached) to remove the basin-scale dependency of the data (based on comments of Reviewer 3, see above).

Page 11, line 9: "Q10" should be "Q90"?

Page 11, line 11: "to a large extent"

Page 15, line 11: Mediterranean should be "(-16%)" not "(-24%)", reading off Table 3 for 2K to 3K?

Page 17, line 1 (and throughout): "Targus" should be "Tagus"?

Table 4: It's more editorial, but Fig 6 discussed before Table 4 despite being featured afterwards.

---

## Author Comment (AC3) · 1 Nov 2017

[Figure]

**Fig. 1.** Scatter plots and spatial distribution between observed and GCM-HM simulated low flow (Q90) over 357 gauges for single HMs and the HM ensemble median for the historical time-period 1965-1995.

---

## Author Comment (AC4) · 1 Nov 2017

author comments RC3 11

November 1, 2017

We thank the reviewer for the time and effort in commenting on our manuscript. We provide responses to each individual point below. For clarity, comments are given in normal font, and our responses are given as blue text.
For newly produced figure 1, please refer to AC3 under reviewer 2.

This manuscript deals with a multi-GCM and multi-hydrological models assessment of changes in low flows across Europe between a present-day period (1971-200) and 3 different global warming levels: 1.5K, 2K, and 3K (and between them as well). It therefore contributes to document the effects of climate change on low-flow hydrology in Europe in the context of the Paris Agreement. This manuscript thus deals with a topical and important topic, and fits well into the scope of HESS. It is generally well structured and written, and conclusions are generally well supported by results shown. I have however two main comments (as well as specific comments) detailed below that should be addressed before the manuscript is published in HESS.
Thank you very much for the extensive commenting and feedback which will help to increase the quality of the manuscript.

**1 Main comments**

**1.1** Hydrological calibration and simulation over influenced catchments The calibration details (specific comment #9 and #10) as well as the validation results (specific comments #12, #13, #14) do not give enough confidence on the quality of hydrological modeling, and highlights the issue of calibrating and/or validating – seemingly natural-catchment-only – models against highly influenced catchments like the Ebro or the Rhône, especially for low flows. First there is not enough information on the calibration process, and even the catchments used for that are not identified. Second, validation is done for a large part over influenced catchments, and also over ensembles of highly nested catchments. Both points should be reconsidered in a future revision of the manuscript.

We acknowledge these facts. The information given on calibration and validation will be modified and extended in the manuscript. More information is given in the specific comments.

**1.2** Scale of catchments selected for presenting and averaging results There are numerous inconsistencies throughout the manuscript in terms of the minimum catchment size used for presenting results (and giving averaged figures), see specific comments #20, #21, #22, #24, #27. Addressing this comment may imply reformatting all results, but this is also intrinsically linked to main comment #1. Indeed, the manuscript state that the runoff routing scheme prevent using results for catch- ments smaller than 10000 km2, and near-natural catchments are usually only smaller than that in Europe. In parallel, maps of results are given over a river network en- compassing drained areas much small than the indicated threshold. This thus shades doubts (maybe unjustified, but is has to be demonstrated) on the validity of models and results, together with issues highlighted in main comment #1.

The information on catchment sizes will be extended in the manuscript. Smaller catchments <10000 km$^2$ have not been omitted. The catchment size at a spatial resolution of $5 \times 5$ km $^2$ is limited e.g. by the DEM in determining the catchment boundaries. Therefore, results from catchments (or better river grid cells) with a contributing area >1000 km$^2$ have been used in the study, and these are shown in
figure 3 and figure 5 and have been used for drawing conclusions. The selection of catchments >10000 km$^2$ in figure 1 and 4 has been done for clarity reasons (clearness of the figures) only. This will be changed for validation figure 1. More information is given in the specific comments.

**2 Specific comments**

**1.** P1L2, "1.5, 2 and 3 K": please specify that this is with respect to the preindustrial period
Agreed.
**2.** P1L10, "-12%": What is the baseline period here? This is all the more important that there could easily be confusion with the baseline used for the global warming level (see above).
Agreed. Baseline period for determining relative changes is 1971-2000. This information will be included.
**3.** L11-12: this sentence is ambiguous. Less snowmelt may imply less streamflow in some conditions (e.g. constant liquid precipitation or declining total precipitation). Please rephrase and make it clearer.
Agreed. We will modify the text as suggested.
**4.** P1L13-14: This sentence is also quite ambiguous. What is exactly preventing distinguishing between 1.5 and 2K warming effects? Is it the interannual variability which prevents distinguishing statistically estimates of period-averaged changes- for a given GCM-HM combination? Or is it the uncertainty due to the multimodel ensemble that prevents distinguishing ensembles of multimodel period-average estimates between present and future? Or both? Please make it clear here.
Agreed. It is both and we will include the information on uncertainty.
**5.** P2L11: The low-flow component of the 2015 drought event has been specifically studied by Laaha et al. (2017). I believe this reference is worth adding to the

manuscript.

*Agreed.*

**6.** P2L21-27: What is the time slice that corresponds to the quantitative and qualitative results recalled here? Please make it clearer.

*1971-200. We sugegst to phrase "until the end of the century."*

**7.** P4L7-10: First, this interpolation step should not be called downscaling as the latter refers to methods that actually add information for each day (either through regional climate models or empirical-statistical downscaling models) to the larger scale GCM fields. I would therefore strongly recommend using "disaggregating" or "disaggregated" instead of "downscaling" or "downscaled" in the manuscript.

*We will modify the text as suggested.*

**8.** P4L7-10: Second, this interpolation step should be better documented here, in order for the reader to understand the advantages and shortcomings of this approach, which are essential for assessing the quality of subsequent hydrological simulations. This interpolation step should ideally be assessed using a global reanalysis against high-resolution gridded datasets, like RCMs are actually as- sessed (see e.g. Kotlarski et al., 2017, among many others, for a recent exam- ple). This would critically allow distinguishing errors coming from (1) the spatial interpolation technique (and their large-scale forcings), and (2) the hydrological models. Please at least add some comments on that in the manuscript. Plus, the reference used for this interpolation technique is incomplete in the list of ref- erences.

*We will extend the text as suggested. Assessing the meteorological input fields against other sources is out of the scope of this study. The missing reference will be added.*

**9.** P4L24: What are these 9 catchments? Please provide some more information (location, surface, etc.). Are they near-natural or influenced catchments?

*#9, #10 and #11 are commented together under #11*

**10.** P4L24-26: What is the period used for calibration? And what are the calibration criteria (for both automatic and manual calibration)? Are they specific for low flows? Please carefully specify all this in the manuscript.

**9, #10 and #11 are commented together under #11**

**11.** P4L27-29, "The assessment ... (Gosling et al., 2017)": This is a very strong statement, which I tend to disagree with at least as a general conclusion. This is moreover hardly supported by the reference given in the manuscript, which compares global hydrological models and catchment hydrological models for the Rhine and Tagus (and other catchments, but not located in Europe). Results for a low flow indicator (Q95) show a large divergence of the two types of models with increasing global warming level (Gosling et al., 2017, their Fig. 2). As a conclusion, I would therefore strongly recommend removing this statement from the manuscript.

Comments #9, #10 and #11 are interrelated and are answered connectedly.

It is important to recognise that all HMs applied are well-established, widely applied and have been used (and calibrated) for Europe in former studies referred to in the manuscript. Furthermore, additional calibration for the three HMs was done in focus basins. Nevertheless, the validity of calibrated parameters may be limited in CC studies (**?**) and the results in multi-HM climate impact studies may be less influenced by the calibration than by the model-structure of the HMs. This could be shown e.g. in **?**.

We agree to #16 that the statement "The assessment ... is independent" is too strong, we would remove this statement with the citation Gosling 2017 and replace it with "Well-calibrated HMs do not necessarily mean that future discharge under a changed climate can be reproduced satisfactorily (**?**). Furthermore, the selection of HMs may have a larger effects than calibration in hydrological climate impact studies (**?**)".

Considering the HMs calibration (#9 and #10) we suggest to extend the manuscript and include the paragraph: "Furthermore, the three HMs used in this study were calibrated in nine near-natural European focus basins located in Spain, Norway and UK, which were selected based on the consultation with the user groups within the EDgE project. Besides these, we also included three more central EU catchments (located in France and Germany) to represent diversity in hydro-climatic regimes. All HMs parameters were calibrated such that the model simulations represent a range

of hydrologic regimes, rather than tailored to any specific characteristics. This was done in a consistent manner so that the model simulations can be used for a range of indicators (including high, low, and average flows) within the EDgE project. HMs could be calibrated to specific parts of the flow duration curve (FDC), however, this was not done in this study to avoid too specific tuning of the model simulations to those unique conditions and thereby losing valuable information on the entire FDC.

In the current simulations human water management was not taken into account, since some models lack the ability to include these processes and one focus on this work is on determining the HM uncertainty in low flow conditions. Human water management can however have a significance impact on the low flow conditions, due to abstraction of additional water in drought conditions or changes in reservoir management - as a result constraining the model to any specific low flow characteristic can result in a biased simulations. Also due to the similar reason we may expect a relatively lower model skill in matching the observed low flow characteristic."

**12.** P4L33-P5L4 and Figure 1: The assessment of HMs is very light and not strongly supported by Fig. 1. Indeed, this figure is potentially misleading, as it basically only checks that catchments have equally small/large indicators (Q90 or Q50) for both observations and simulations, which is mainly driven by the size of the catchment. I would therefore recommend using a different and more informative representation of differences, preferably in terms of relative errors (in percents), and also preferably as maps in order to show the potential spatial pattern in errors. This representation would also greatly help in comparing present-day errors to relative changes presented later in the manuscript. I personally would not give too much credit for a model showing for a given location present-day errors as large as 3K future changes...

We agree to change the metric to remove the catchment size effect. Therefore, we would use specific discharge [mm/d], include the information on the HM ensemble mean, and show the relative bias spatially distributed (see attached figure). We tend to disagree with the statement " I personally would not give too much credit for a model showing for a given location present-day errors as large as 3K future changes...".

The comparison shown here is an "honest" one because the HMs are driven with GCM input for a time period in the past. This is usually not shown in climate impact studies. Furthermore, assuming a constant error or bias over time in the GCM-HM simulations would result in perfect study results. Therefore, we trust the uncertainty measures presented in this study (SNR combined with robustness) more. Considering the relative biases shown in the attached figure it is important to notice that the spatial pattern is very different from the climate change signal (Fig. 3 and 5). It would be critical if these patterns would match.

**13.** Figure2, right: This figure shows the location of validation gauges used in Fig. 1. First, it shows that many points in Fig. 1 comes from the same rivers and are necessarily highly correlated, which inherently bring some bias to the results that should be representative of the whole Europe. I would strongly recommend removing redundant points scatter plots like presented in Fig. 1. This would not be a problem however with suggested spatial representations (cf. above).

The validation gauges have been changed and more gauges are included in the revised manuscript. This will be changed (attached figure ), and additionally, spatial representations are shown.

**14.** Figure 2, right: The second point is that several validation gauges are located on highly influenced rivers. For example, the Ebro river (Spain) is heavily influenced by water abstractions for irrigation, and the seasonal regime of the Rhône river (France) is heavily influenced by all the hydropower reservoirs located in the Alps (and other surrounding mountain ranges). There are many other cases that can be spotted on the map. As a consequence, observed streamflow indicators for low flows simply cannot be compared to natural (i.e. without human influence) hydrological simulations for these catchments. A good fit to observations may indeed reveal that physical parameters in HMs are tweaked to compensate for no representation of human influence. This may not be a problem in itself (at least for practical modeling purposes if not scientifically satisfactory) if human influences would not have changed and would not change in the future. Which has happened and definitely will. As a conclusion,

I would strongly recommend using only near-natural catchments as validation (and also calibration) gauges for natural hydrological modeling (as I suppose it is the case in the manuscript, even if some HMs considered may represent human influences). A number of reference hydrometric networks have recently been developed at the country scale (Hannaford and Marsh, 2008; Giuntoli et al., 2013; Murphy et al., 2013), and one should take advantage of these. Note that these networks overlap for some countries (but not for some other) with stations tagged "climate sensitive" in the Global Runoff data Centre.

Calibration in HMs has been performed using headwater catchments and no heavily human influenced basin was included. It would generally be a good idea to use "climate sensitive" stations only. Unluckily, these are not uniformly distributed all over Europe, but only available in selected countries. Esp. in the Mediterranean area there is no such station available. We would consider this comment in future studies in case an area-wide coverage of climate sensitive stations is available.

**15.** P6L3: The 0.46K figure has uncertainties attached to it, according to the reference cited (Vautard et al., 2014). Please do mention these uncertainties in the manuscript, with possibly additional references that provides 1971-2000 estimates of global warming level.

Agreed. "The warming of 0.46 K in an average value from three estimations with a spread between 0.437 K and 0.477 K."

**16.** P6L20: The use of calendar year is not entirely satisfactory for computing Q90 in snow-influenced catchments where the low-flow period (or one of the low-flow periods, which is a more difficult situation) may span two calendar years. Please consider changing the calculation procedure or at least justify this approximation.

We will extend the text including the limitation mentioned.

**17.** P7L8-9: Please mention here (rather than in the results section) that the robustness is compute as the percentage of projections showing a significant change.

Agreed.

**18.** Table 2: Please make clear that "1980s" refers to the 1971-2000 period.

Agreed.

**19.** P8L3-9: I don't really understand this peculiar choice of method for computing the relative contributions of uncertainty from GCMs and HMs. Many studies demonstrated that simple Analysis of Variance (ANOVA) approaches are perfectly suited to this case, and it has been recently widely applied to compute contribution from GCMs and HMs (see e.g. Giuntoli et al., 2015; Vetter et al., 2017, among many others), even by some of the authors of the present manuscript (Mishra et al., 2015). ANOVA approaches can critically take account of GCM/HMs interactions, which is presumably not the case of the method used, and of the different sizes of fixed effects. The set-up is here rather simple compared to more complex ones that consider unbalanced number of runs from each GCMs and/or multiple sources of uncertainty (see e.g. Addor et al., 2014; Vidal et al., 2016). I therefore strongly recommend using a simple two-way ANOVA approaches for the present study, or at least check current results against a simple two-way ANOVA approach. Indeed, I am unsure of how this sequential sampling approach relates to the more traditional ANOVA approach, and what their respective underlying hypotheses are. I would welcome some online discussion on this.

The rationale of not using ANOVA and the description of the sequential sampling procedure similar to that proposed by (**?**) was explained in (**?**). In short, standard parametric procedures, such as the Analysis of Variance (ANOVA), require assumptions of normality to estimate significance levels (the F and the t-student test require that the underlying variable is normally distributed). The low-flow statistics estimated in this study are non-normal and hence standard methods are not appropriate. For the estimation of the relative contributions of uncertainty from GCMs and HMs we use the range of the ensemble instead of the variance as suggested by Schewe et al. The confidence interval and the significance level of variability is estimated, in this case, with a non-parametric method (basically it is bootstrapping). This method is called sequential sampling in Samaniego et al. 2016. because it includes a "sampling with replacement" procedure to generate confidence intervals for the range statistic. Moreover, a non- parametric (bootstrapping) procedure is preferred here to reduce the

effects of the biased variance estimation due to the small sample size.

**20.** P8L13-15: First, this should come much earlier in the manuscript. Second, this is not consistent with maps of streamflow changes that seemingly include results for catchments with a surface lower than 10000 km2. This should be clarified. This is closely linked to specific comment #14.

First: Agreed. Second: explained in 1.2. The information that river grid cells with a contributing area >1000 km$^2$ are used in the results section will be included.

**21.** Figure 3 (and Fig. 5 and Fig. 6). See comment above. Plus, the figure indicated above each map is seemingly a continental average of the plotted value along the river network. First, this should be clarified. Second, this value is closely related to the choic3-15, e of the minimal catchment surface area considered. Values would be very different if, as stated P8L1only catchments with an area larger than 10000 km2 would be considered. Please make all these statement and results consistent across the manuscript.

Agreed.

**22.** P10L3-4: This statement is somewhat inconsistent with the choice of the calendar year use for the calculation of Q90. Please clarify this in the manuscript.

Agreed. See #16

**23.** P10L7, "models": I presume this should be "simulations".

Yes, right.

**24.** P11L1, "new spatially explicit information". This is again contradictory with the 10000 km2 statement. Cf. comments above.

See general comment 1.2

**25.** P11L16-17.This sentence is ambiguous. The increased spread along the 1:1 line (i.e. when smaller and larger values are considered) does indeed contribute to a higher coefficient of determination, which is not the case for the spread across (i.e. with higher residuals from) the 1:1 line. Please rephrase.

Agreed.

**26.** Figure 4: Several presumably regression lines are given on the graph. Please

either define and comment them, or remove them. Also, please add lines delimiting the quadrants.

The regression lines are shown for positive and negative values. Modifications will be done as proposed.

**27.** Figure 4: The legend states that only catchments with a surface area higher than 10000 km2 are considered. This is again not consistent with values provided by other figures.

This is true and was done with respect to the clearness of the figure. A comparison to surface areas higher than 1000 km$^2$ showed similar results.

**28.** P13L3-5: This is already written P10L35-P11L2. And this is commented in specific comment #24.

See #24

**29.** Title of Section 3.2: The difference between section 3.1 and section 3.2 are not understandable based on this title, and the reader may be unsettled at this point as I was. There should be something of a "between the levels of warming" somewhere. Please rephrase.

Agreed.

**30.** Figure 5. Cf. comment #21.

Agreed.

**31.** P15L15-16: The increase in winter low flows would not necessarily lead to a higher hydropower potential. It actually depends on the evolution of total precipitation. And the possible evolution of hydropower production would depend on the type of reservoir management, as well as management rules constrained by possible other water usages (sustaining summer low flows downstream, irrigation, recreation, etc.). Moreover, a decrease in low flows does not necessarily imply a decrease in overall water availability average over the year, and the water stress is conditional on the respective weight of water availability and water demand for a given time. So I would recommend adapting the statements according to the above comments.

Agreed.

**32.** P15L17-18: I however completely agree with the need of regional adaptation options. Except that adaptation strategies should be put in place now, without waiting for the 3K level to be reached or not.

Agreed. We will include a sentence on this.

**33.** P16L6, "the result is independent of the sign of change": Well, this is a potentially serious issue. Indeed, how to interpret a situation where e.g. out of15 projections, 5 give a significant upward change, 5 other no significant change, and the last 5 a significant downward change? I would recommend interpreting this situation with particularly no robust signal! So please make clearer in the manuscript all the different possible cases and the way to interpret them. An alternative for presenting robustness would be the one used in the IPCC AR5 WGI report, i.e. the percentage of projections agreeing on the sign of the change.

Fully agree with first statement. This is the reason why we suggest to use a combination of SNR and robustness. E.g. using the IPCC AR5 WGI report would give an information similar to SNR, but without a significance information.

**34.** P16L11-13: I totally agree with this sentence, but it comes here out of the blue. Please consider moving it to the introduction, discussion, or conclusion.

The sentence will be added in the conclusions.

**35.** Figure 6. The choice of colour breaks is here particularly unfortunate here. For the SNR, I would appreciate having a break in value 1, in order to see where the median change is higher than the uncertainty in projections. For the ratio of GCM to HM uncertainty contribution, this is all the more important to see where this crosses the 1 value. An alternative would be to use bivariate colour scales (Teuling et al., 2011) to jointly plot the evolution of both sources of uncertainty.

We understand that it is naturally to expect color breaks at 1 and we also used these at a previous version of this Figure. The purpose of this section is to highlight the substantial uncertainties associated with the results. For this reason, we decided to use a range of plus/minus 20% around 1. to mark regions where the contribution by GCMs and HMs (GCM/HM contr.) are of the same order of magnitude (please note

that 0.8 and 1.25 are the inverse of each other for multiplication). It is measleading to distinguish a value slightly higher than 1. from one slightly lower (e.g., 1.04 from 0.98). Given the uncertainty in the analysed dataset, we only consider a higher contribution by either GCMs or HMs if it is at least 20% higher than the other. Similar arguments hold for the signal to noise ratio. We added a sentence to clarify these points (see p. 15 line 10). Using a bivariate color scheme following Teuling et al. 2011 is a possible alternative for the presentation of Figure 6. This color scheme would show the values in absolute terms rather then relative to each other. It would be possible to distinguish high and low values, but it would be harder to see which of the two sources of uncertainty is higher. We also think that showing the signal to noise ratio already allows to identify regions with high and low uncertainty and, additionally, providing absolute values is not required. Proposed paragraph to include: "It is worth noting that we have chosen the color scheme in Figure 6 in a way that regions where the SNR is within 20% around 1. have the same colors. These regions have a signal, which is of similar magnitude as the uncertainty. Different colors are used to mark regions where the the signal is more than 20% higher or lower than the uncertainty."

**36.** P17L4-5: This exact sentence has already been written P8L3-4, and commented above (comment #19)

The sentence will be rephrased.

**37.** P17L8-P18L3: I am more or less OK with what is written here, but I do not understand why this would imply that the ratio of HM contribution to GCM contribution is higher at the 3K level. Please provide some explanations in the manuscript. Couldn't this be related to timing of threshold crossing in HM behavior that would differ from one HM to another, e.g. going from energy-limited to water-limited evaporation process?

Answered under #38

**38.** P18L4-20: This whole paragraph tends to support the above hypothesis. This should be related in the manuscript to recent uncertainty decomposition results obtained for a catchment located in the Southern Alps. It showed that the increasing spread of changes in future low flows by different HMs is linked to increasing spread

in simulated evaporation and snow water equivalent (Vidal et al., 2016).

We agree with the reviewer that the explanation that we provided is only valid for the increase of the uncertainty of GCMs and that other factors such as the one mentioned by the reviewer influence the increase of HM uncertainty. We rephrased this paragraph to be more explicit about the different sources influencing the uncertainty contribution paragraph starting at p. 17, l. 7:

Total uncertainties in low flow projections is separated into GCM and HM contributions using the sequential sampling method proposed in Schewe et al. (2014). The results are shown in Fig. 6 (d-f) and spatially aggregated over the IPCC Europe regions in Tab. 4. The uncertainty rises with higher levels of warming for both sources of uncertainty because of two reasons. The GCM uncertainty increases because a 30 year period reaching a 3 K warming often has a strong temperature period within this 30 year period. Contrarily, GCM runs under RCP 2.6 often stabilise around 1.5 K global warming. This pathway dependency of warming influences the variability of the results with expectedly higher variability in the former case (James et al., 2017). The HM uncertainty increase with global warming because certain regions might crosse thresholds. For example, parts of France might move from a energy-limited to a water-limited regime. Overall, the contribution of the GCMs to the uncertainty over Europe is about 21% higher under 1.5 K, 25% higher 10 under 2 K and only 10% higher under 3 K global warming in comparison to the HM contribution. This decrease of GCM/HM contribution can be mostly attributed to the Mediterranean and Atlantic region (in particular France). In these dry regions, the different representations of evaporation using temperature-based potential evapotranspiration used in mHM and PCR-GLOBWB lead to different responses than explicitly solving the full energy-balance of the land surface as in Noah-MP.

**39.** P19L28-30, "We conclude. . . support the adaptation process." Well, this is actually only a wish. Nothing in the paper allows asserting that, even I personally hope this is the case. So please rephrase.

Agreed.

[Figure]

**3 Technical corrections**

Technical corrections hereafter will be adressed according to the reviewers suggestions.

1. P1L5, "unprecedented": it is a bit far-fetched, given that (1) GCM forcings are only disaggregated to this resolution without adding any downscaling information, and (2) results are seemingly partly given only for catchments >10000 km2 (P8L13- 15).
2. P1L6: "combination"
3. P2L2: "independently"?
4. P2L22-24: I believe that the sentence is not grammatically correct.
5. P2L30: "2"? in reference (UNFCC, 2015)
6. P2L34: "because of"
7. P3L3: please check missing or incorrect "the"
8. P3L8: "southern Europe"
9. P11L11: "extent"
10. P13L5: "political" -> "policy". Also P15L23.
11. P15L22, "distinguished": please rephrase.
12. P15L24, "ensemble members": Please clarify what they are.
13. P20L1, "pronounced": What is? Please rephrase.
14. P21L5-8: Wrong formatting, cf. IPCC report citation rules.
15. P23L34: line feed
16. P24L25-26: extra information to be removed

**4 References**

Mendoza, P. A., Clark, M. P., Mizukami, N., Newman, A. J., Barlage, M., Gutmann,

HESSD
E. D., Rasmussen, R. M., Rajagopalan, B., Brekke,L. D., and Arnold, J. R.: Effects of Hydrologic Model Choice and Calibration on the Portrayal of Climate Change Impacts, Journal of Hydrometeorology, 16, 762–780, doi:10.1175/JHM-D-14-0104.1, https://doi.org/10.1175/JHM-D-14-0104.1, 2015.

Samaniego, L., Kumar, R., Breuer, L., Chamorro, A., Flörke, 5 M., Pechlivanidis, I. G., Schäfer, D., Shah, H., Vetter, T., Wortmann, M., and Zeng, X.: Propagation of forcing and model uncertainties on to hydrological drought characteristics in a multi-model centurylong experiment in large river basins, Climatic Change, 141, 435–449, doi:10.1007/s10584-016-1778-y, http://dx.doi.org/10.1007/s10584-016-1778-y, 2017.

Schewe, J., Heinke, J., Gerten, D., Haddeland, I., Arnell, N. W., Clark, D. B., Dankers, R., Eisner, S., Fekete, B. M., Colón-González, F. J., Gosling, S. N., Kim, H., Liu, X.,Masaki, Y., Portmann, F. T., Satoh, Y., Stacke, T., Tang, Q.,Wada, Y.,Wisser, D., Albrecht, T., Frieler, K., Piontek, F., Warszawski, L., and Kabat, P.: Multimodel assessment of water scarcity under climate change, Proceedings of the National Academy of Sciences, 111, 3245–3250, doi:10.1073/pnas.1222460110, http://www.pnas.org/content/111/9/3245.abstract, 2014.

Vaze, J., Post, D., Chiew, F., Perraud, J.-M., Viney, N., and Teng, J.: Climate non-stationarity – Validity of calibrated rainfall–runoff models for use in climate change studies, Journal of Hydrology, 394, 447 – 457, doi:https://doi.org/10.1016/j.jhydrol.2010.09.018, http://www.sciencedirect.com/science/article/pii/S0022169410005986, 2010.